# Polyols and glucose particulate species as tracers of primary biogenic organic aerosols at 28 French sites

Abdoulaye Samake[1], Jean-Luc Jaffrezo[1], Olivier Favez[2], Samuël Weber[1], Véronique Jacob[1], Alexandre Albinet[2], Véronique Riffault[3], Esperanza Perdrix[3], Antoine Waked[1], Benjamin Golly[1], Dalia Salameh[1†], Florie Chevrier[1], Diogo Miguel Oliveira[2,3], Nicolas Bonnaire[4], Jean-Luc Besombes[5], Jean M.F Martins[1], Sébastien Conil[6], Géraldine Guillaud[7], Boualem Mesbah[8], Benoit Rocq[9], Pierre-Yves Robic[10], Agnès Hulin[11], Sébastien Le Meur[12], Maxence Descheemaecker[13], Eve Chretien[14], Nicolas Marchand[15], and Gaëlle Uzu[1].

[1]Univ. Grenoble Alpes, CNRS, IRD, INP-G, IGE (UMR 5001), F-38000 Grenoble, France.
[2]INERIS, Parc Technologique Alata, BP 2, 60550 Verneuil-en-Halatte, France
[3]IMT Lille Douai, Univ. Lille, SAGE – Département Sciences de l'Atmosphère et Génie de l'Environnement, F-59000 Lille, France
[4]LSCE, UMR CNRS-CEA-UVSQ, 91191 Gif-sur Yvette, France
[5]Univ. Savoie Mont-Blanc, LCME, F-73000 Chambéry, France
[6]ANDRA DRD/GES Observatoire Pérenne de l'Environnement, F-55290 Bure, France
[7]Atmo AuRA, F-38400 Grenoble, France
[8]Air PACA, F-03040, France
[9]Atmo Hauts de France, F-59000, France
[10]Atmo Occitanie, F-31330 Toulouse, France
[11]Atmo Nouvelle Aquitaine, F-33000, France
[12]Atmo Normandie, F-76000, France
[13]Lig'Air, F-45590 Saint-Cyr-en-Val, France
[14] Atmo Grand Est, F-16034 Strasbourg, France
[15]Univ Aix Marseille, LCE (UMR7376), Marseille, France
[†]Now at Airport pollution control authority (ACNUSA), 75007 Paris, France

Corresponding author(s): A Samaké (abdoulaye.samake2@univ-grenoble-alpes.fr) and JL Jaffrezo (Jean-luc.Jaffrezo@univ-grenoble-alpes.fr)

**Abstract.** A growing number of studies is using specific primary sugar species, such as sugar alcohols or primary saccharides, as marker compounds to characterize and apportion primary biogenic organic aerosols (PBOA) in the atmosphere. To better understand their annual cycles, as well as their spatio-temporal abundance in terms of concentrations and sources, we conducted a large study focusing on three major atmospheric primary sugar compounds (i.e., arabitol, mannitol and glucose) measured in various environmental conditions on about 5,300 filter samples collected at 28 sites in France. Our results show significant atmospheric concentrations of polyols (defined here as the sum of arabitol and mannitol) and glucose at each sampling location, highlighting their ubiquity. Results also confirm that polyols and glucose are mainly associated with the coarse rather than the fine aerosol mode. At nearly all sites, atmospheric concentrations of polyols and glucose display a well-marked seasonal pattern, with maximum concentrations from late spring to early autumn, followed by an abrupt decrease in late autumn, and a minimum concentration during wintertime. Such seasonal patterns support biogenic emissions associated with higher biological metabolic activities (e.g., sporulation, growth, etc.) during warmer periods. Results from a previous comprehensive study using Positive Matrix Factorization (PMF) based on an extended aerosol chemical composition dataset of up to 130 species for 16 of the same sample series has also been used in the present work. The Polyols-to-$PM_{PBOA}$ ratio is $0.024\pm0.010$ on average for all sites, with no clear distinction between traffic, urban or rural typology. Overall, even if the exact origin of the PBOA source is still under investigation, it appears to be an important source of PM, especially during summertime. Results also show that PBOA are significant sources of total OM in $PM_{10}$ ($13\pm4$ % on a yearly average, and up to 40 % in some environments in summer) at most of the investigated sites. The mean PBOA chemical profile is clearly dominated by contribution from organic matter (OM) ($78\pm9$ % of the mass of the PBOA PMF factor on average), and only a minor contribution from dust class ($3\pm4$ %), suggesting that ambient polyols are most likely associated with biological particle emissions (e.g., active spore discharge) rather than soil dust resuspension.

## 1. Introduction

Airborne particles (or particulate matter, PM) are of major concern due to their multiple effects on climate and adverse human health impacts (Boucher et al., 2013; Cho et al., 2005; Ntziachristos et al., 2007). The diversity of PM impacts is closely linked to their complex and highly variable nature: size distribution, concentration and chemical composition, or specific surface properties. PM consists of inorganic and elemental substances, and a large fraction made of carbonaceous matter (organic carbon (OC) and elemental carbon (EC)) (Franke et al., 2017; Putaud et al., 2004a; Yttri et al., 2007a). Substantial amounts of atmospheric organic matter (OM) remain unidentified and uncharacterized at the molecular level. In most studies, a maximum of only 20 % of particulate OM mass can generally be speciated and quantified (Alfarra et al., 2007; Fortenberry et al., 2018; Liang et al., 2017; Nozière et al., 2015). This detailed composition of OM and its spatial and seasonal distribution can give important insights on the adverse effects of PM. So far, the majority of air pollution studies have focused on organic atmospheric particles associated with anthropogenic and secondary sources, whereas a significant fraction of OM can also be associated with primary emissions from biogenic sources (Bauer et al., 2008a; Jaenicke, 2005; Liang et al., 2016). Therefore, the characterization of primary OM biogenic sources at the molecular level is still limited (Fuzzi et al., 2006; Liang et al., 2017; Zhu et al., 2015), and should be further investigated for a better understanding of aerosol sources and formation processes.

Primary biogenic organic aerosols (PBOA) are emitted directly from the biosphere to the atmosphere where they
are ubiquitous and participate in many atmospheric processes (Elbert et al., 2007; Fröhlich-Nowoisky et al., 2016).
Additionally, their inhalation has long been associated with human respiratory impairments (e.g., asthma,
aspergillosis, etc.) (Després et al., 2012; Morris et al., 2011). PBOA comprise living and dead microorganisms
such as bacteria, fungi, viruses, bacterial and fungal spores, and microbial fragments, endotoxins, mycotoxins, or
pollens (Elbert et al., 2007; Jaenicke, 2005; Morris et al., 2011). In most semi-urban European sites, PBOA can
account for up to  25 % of the atmospheric aerosol mass, in the size range of 0.2 to 50 µm (Fröhlich-Nowoisky et
al., 2016; Jaenicke, 2005; Huffman et al., 2012; Manninen et al., 2014; Morris et al., 2011). However, their sources
and contribution to total airborne particles are still poorly documented, partly because of the difficulty to recognize
them by conventional microbiological methods (cell culture, microscopic examination, etc.) (Di Filippo et al., 2013;
Heald and Spracklen, 2009; Jia et al., 2010a).
Several specific chemical components, such as primary sugar compounds (i.e., primary saccharides and sugar
alcohols) emitted persistently from biogenic sources, have long been suggested as powerful and unique biomarkers
in tracing sources, and abundances of PBOA as well (Bauer et al., 2008a; Medeiros et  al., 2006; Simoneit et al.,
2004b; Zhang et al., 2010; Zhu et al., 2016). For instance, ambient concentrations of glucose have been used as
markers for plant materials (such as pollen, leaves, and their fragments) or soil emissions from several areas in the
world (Fu et al., 2012; Jia et al., 2010a, 2010b; Pietrogrande et al., 2014; Rathnayake et al., 2017). Many studies
indicated that glucose is the most abundant monosaccharide in vascular plants, where it serves as the common
energy material, and an important source of carbon for soil active microorganisms (such as bacteria or fungi) (Jia
et al., 2010a; Medeiros et al., 2006; Pietrogrande et al., 2014; Zhu et al., 2015). Additionally, sugar alcohols (also
called polyols) including arabitol and mannitol have been proposed as markers for airborne fungi, and are widely
used to quantify their contributions to PBOA  mass (Bauer et al., 2008a, 2008b; Golly et al., 2018; Srivastava et
al., 2018; Zhang et al., 2010). These sugar alcohols have also been found to correlate very well with fluorescent
PBOA in the ultraviolet aerodynamic particle sizer (UV-APS) and wideband integrated bioaerosol sensor (WIBS-
3) online studies, particularly in rainy periods (Gosselin et al., 2016), favoring microbial sporulation (such as fungi
belonging to Ascomycota and Basidiomycota phyla) (China et al., 2016; Elbert et al., 2007; Jones and Harrison,
2004). Polyols are produced in large amounts by many fungi and bacteria, and several functions have been
described for these compounds, such as common energy storage materials, intracellular protectants against
stressful conditions (e.g., heat or drought), storage or transport of carbohydrates, quencher of oxygenated reactive
species, or regulators of intracellular pH by acting as a sink or source of protons (Jennings et al., 1998; Medeiros
et al., 2006; Vélëz et al., 2007). Hence, polyols, especially arabitol and mannitol, may represent a significant
fraction of the dry weight of fungi, and mannitol can contribute between 20 to 50 % of the mycelium dry weight
(Ruijter et al., 2003; Vélëz et al., 2007). However, polyols are also often identified in the lower plants (leaves,
pollens) and green algal lichens (Medeiros et al., 2006; Vélëz et al., 2007; Yang et al., 2012). The primary sugar
compounds (defined as polyols and primary saccharide species) are thought to be relatively stable in the
atmosphere (Wang et al., 2018), although studies investigating their atmospheric lifetime are quite limited. One
previous laboratory study has been conducted by the US-EPA to evaluate the stability of these chemicals on filter
material exposed to gaseous oxidants as well as in aqueous solutions (simulating clouds and fog droplet chemistry).
Findings of this former study have shown that primary sugar compounds remain quite stable up to 7 days (the
extent of the testing period), pointing out their suitability for use as tracers of atmospheric transport (Fraser, 2010).
With all of this information, the use of primary sugar compounds (such as mannitol, arabitol, glucose, etc.) as
suitable tracers of PBOA is generally acknowledged (Jia and Fraser, 2011; Zhu et al., 2015, 2016).
Although atmospheric concentrations of polyols, including arabitol and mannitol, as well as that of some primary
monosaccharides (e.g., glucose), have been previously quantified as part of several studies in various environments
including urban/suburban, rural, rainforest, mountain, and marine areas (Fu et al., 2012; Graham et al., 2003; Jia
et al., 2010a, 2010b; Liang et al., 2016; Pietrogrande et al., 2014; Simoneit, 2004a, 2004b; Verma et al., 2018;
Yttri et al., 2007b; Zhu et al., 2015), large datasets investigating their annual cycles and spatial distributions are
still limited. Such information could give important insights on environmental factors influencing their
atmospheric levels such as climate and biotope, and therefore help to elucidate patterns regarding their major
sources and atmospheric emission pathways. Even if numerous sources and emission mechanisms have been
widely proposed, including among others, metabolic active microbial wet emissions, entrainment of farmland or
natural soils and associated microbiota (Elbert et al., 2007; Fu et al., 2013; Gosselin et al., 2016; Jia et al., 2010a,
2010b; Medeiros et al., 2006; Pietrogrande et al., 2014; Simoneit et al., 2004a, 2004b; Verma et al., 2018; Yttri
et al., 2007b), the dominant atmospheric input processes have not been yet sufficiently elucidated.
In this context, the present study was designed to provide a large overview of the spatial and seasonal variations
of polyols and glucose mass concentrations, as well as their contribution to the aerosol organic mass fraction in
France. To do so, data was collected at many sites in different environments (rural, traffic, urban), in order to
represent various sampling conditions in terms of site typologies and meteorological conditions. Thanks to the
availability of results from an extended Positive Matrix Factorization (PMF) analysis performed for the
corresponding datasets, the overall contributions of the main polyols and glucose emission sources could also be
investigated in light of their spatial patterns. To the best of our knowledge, this is the first study providing such an
extended phenomenology of these compounds over multiple sites with different typologies.
**2. Material and methods**
**2.1 Aerosol sampling**
Ambient aerosol samples considered in the present work come from different research and monitoring programs,
conducted over the last 5 years in France (Figure 1). Each program includes at least one-year of field sampling,
providing a total number of 5,343 daily filter samples available for the sake of the present study. These sites offer
diverse conditions in terms of typologies (i.e., rural, traffic, urban sites, Alpine valley environments, etc.), local
climate and vegetation types and were selected in order to cover the complex and variable national environmental
conditions. These sites are assumed to represent typical environmental conditions in France, and our
observations/and general tendency could therefore be extrapolated to neighboring western European countries
presenting quite homogeneous environmental conditions.

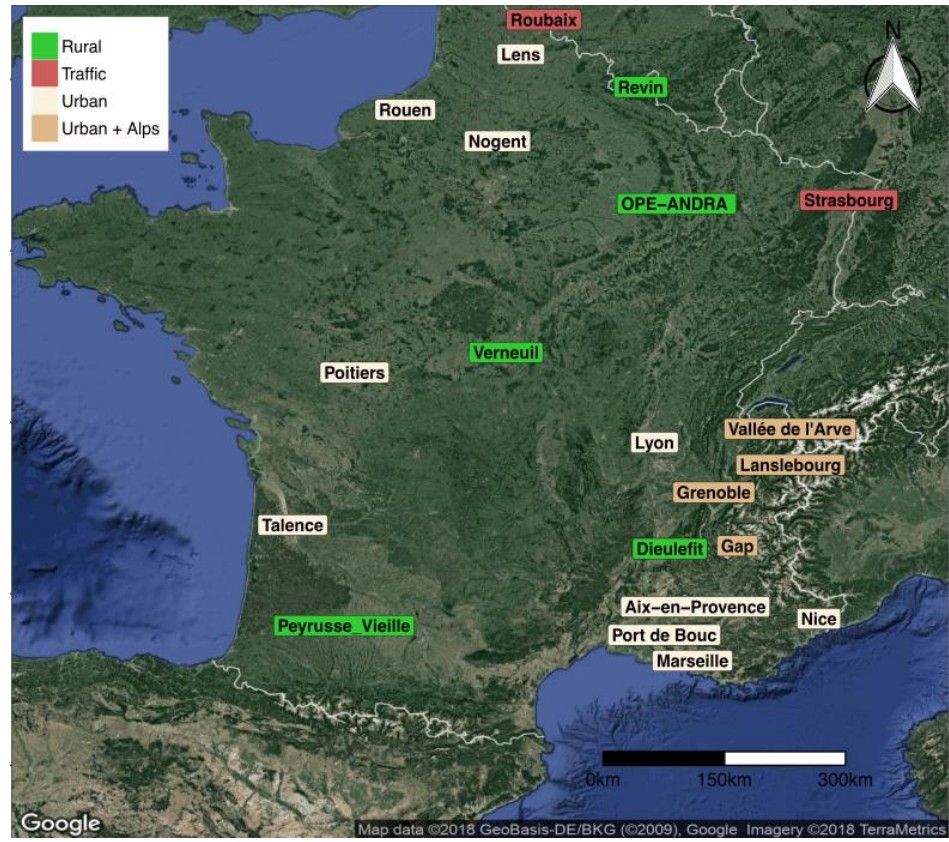


**Figure 1: Map of sampling site locations in France. Green: rural background, red: traffic, wheat: urban background
and dark wheat: urban background in Alpine valley sites. The areas of Grenoble (Grenoble_LF, Grenoble_CB and
Grenoble_VIF) and Vallée de L'Arve (Marnaz, Passy and Chamonix) include 3 sites each. The area of Marseille
includes four sites: Marseille, Mallet, Meyreuil and Gardanne.**

The site of OPE-ANDRA is a specific monitoring observatory in a rural environment, without any village or
industry within several kilometers (description available from: http://www.andra.fr). All other sites correspond to
stations of regional air quality monitoring networks (AASQA). The availability of filter samples was variable from
one site to the other one, depending on the sampling frequency (typically every third or sixth day). Filter collection
was conducted within the $PM_{10}$ or the $PM_{2.5}$ aerosol size fraction, depending on the investigated site (Table SI-1).
Moreover, co-located and simultaneous $PM_{10}$ and $PM_{2.5}$ samplings were conducted at OPE-ANDRA and Revin,
allowing to investigate the distribution of primary sugar compounds between the fine and the coarse aerosol size
fractions at these two sites.

Ambient aerosols were collected onto quartz fiber filters (Tissu-quartz PALL QAT-UP 2500 150 mm diameter),
preheated at 500 °C for 4 h minimum before use. After collection, all filter samples were wrapped in aluminum
foils, sealed in zipper plastic bags and stored at <4°C until further chemical analysis. Field blank filters were also
collected, at least once a month, using the same handling procedure as for PM samples. More detailed information
on the sampling periods, air sampler, number of filters and nature of PM samples are provided in Table SI-1 and
Fig. SI-1.

**2.2 Chemical analyses**

PM samples were analyzed for various chemical species using sub-sampled fractions of the collection filters. In
the frame of the present study, the carbonaceous matter (organic carbon (OC) and elementary carbon (EC)) was
analyzed using a thermo-optical method on a Sunset Lab analyzer (Birch and Cary, 1996) as described by Aymoz

et al. (2007), using the EUSAAR2 temperature program (Cavalli et al., 2010), except for the five sites of Northern
France where the NIOSH870 protocol was employed (Birch and Cary, 1996). OM contents were then estimated
by multiplying the organic carbon mass concentrations by a fixed factor, with $OM = 1.8 \times OC$. This OM-to-OC
ratio value of 1.8 was chosen based on previous studies performed in France (Favez et al., 2010; Petit et al., 2015
and reference therein) and around the world. (e.g., Aiken et al., 2008; Li et al., 2018; Ruthenburg et al., 2014;
Vlachou et al., 2018), with a typical range of 1.2-2.4 values.
For the analysis of anhydrosugars, sugar alcohols, and primary saccharides, filter punches (typically of about 10
cm²) were first extracted into ultrapure water, then filtered using a 0.22 µm Acrodisc filter. Depending on the site,
analyses were conducted either by IGE (Institut des Géosciences de l'Environnement) or by LSCE (Laboratoire
des Sciences du Climat et de l'Environnement) (Table SI-1). At IGE, extraction was performed during 20 min in
a vortex shaker and analyses were achieved using High-Performance Liquid Chromatography (HPLC) with Pulsed
Amperometric Detection. A first set of equipment was used until March 2016, consisting of a Dionex DX500
equipped with three columns Metrosep (Carb 1-Guard + A Supp 15-150 + Carb 1-150), the analytical run being
isocratic with 70 mM sodium hydroxide eluent, followed by a gradient cleaning step with a 120 mM NaOH eluent.
This analytical technique enables to detect anhydrous saccharides (levoglucosan, mannosan, galactosan), polyols
(arabitol, sorbitol, mannitol), and glucose (Waked et al., 2014). A second set of equipment was used after this date,
with a Thermo-Fisher ICS 5000[+] HPLC equipped with 4 mm diameter Metrosep Carb 2 × 150 mm column and
50 mm pre-column. The analytical run is isocratic with 15 % of an eluent of sodium hydroxide (200 mM) and
sodium acetate (4 mM) and 85 % water, at 1 mL min[-1]. This method allows for additional separation and
quantification of erythritol, xylitol, and threalose. At LSCE, extraction was performed during 45 min by sonication
and analyses were achieved using an ion chromatography (IC) instrument (DX600, Dionex) with Pulsed
Amperometric Detection (ICS3000, Dionex). A CarboPAC MA1 column has be used (4 × 250 mm, Dionex), the
analytical run being isocratic with 480 mM sodium hydroxide eluent. This analytical technique enables to detect
anhydrous saccharides (levoglucosan, mannosan, galactosan), polyols (arabitol, mannitol), and glucose.
Field blank filters were handled as real samples for quality assurance. The present data were corrected with field
blanks. The reproducibility of the analysis of primary sugar species (polyols, glucose), estimated from the analysis
of extracts of 10 punches from the same filters is generally in the range of 10-15 %.
Additional chemical analyses were conducted for most of the sites, allowing to quantify up to 130 different
chemical species (Calas et al., 2018). 30- 35 chemical species were then selected in order to achieve PMF analyses
as discussed hereafter.

## 2.3 Statistical analysis

Species concentration measurements were first analyzed for normality using Shapiro-Wilk's method with the
statistical program *R studio interface* (version 3.4.1). Since data were generally not distributed normally, we used
non-parametric statistical methods. The strength of the relationship between species concentrations was
investigated using the non-parametric Spearman rank correlation method. Multiple mean comparison analyses
were performed with the Kruskall-Wallis test method. Statistical significance was set at $p < 0.05$.
Positive Matrix Factorization for the source apportionment of the PM was previously performed at several sites of
this study, as part of the SOURCES (Favez et al., 2016; Salameh et al., in prep.) and DECOMBIO (Chevrier, 2017)
projects. We used the US EPA PMF 5.0 software (US EPA, 2015), following the general recommendation

guidelines of the European Joint Research Centre (JRC) (Belis et al., 2014). Briefly, the SOURCES program aimed at performing source apportionment at 15 sites using a harmonized methodology, i.e., using the same chemical species, uncertainties, constraints, and criteria for factor identification. The PMF conducted within SOURCES project uses about 30 different species (Table SI-6), including carbonaceous fraction (OC, EC), ions ($Cl^-$, $NO_3^-$ $SO_4^{2-}$, $NH_4^+$ $K^+$, $Mg^{2+}$, $Ca^{2+}$), organic markers (polyols i.e., sum of arabitol, mannitol and sorbitol; levoglucosan; mannosan) and metals (Al, As, Ba, Cd, Co, Cs, Cu, Fe, La, Mn, Mo, Ni, Pb, Rb, Sb, Se, Sn, Sr, Ti, V, Zn). The PMF conducted within the DECOMBIO project, for the sites of Marnaz, Chamonix, and Passy, used aethalometer (AE 33) measurements instead of EC (Chevrier, 2016). This complementary measure gives the total black carbon (BC), thus enabling the deconvolution of BC concentrations into its two main constituents: wood-burning BC ($BC_{wb}$) and fossil-fuel BC ($BC_{ff}$) (Sandradewi et al., 2008). For graphical simplicity, $BC_{wb}$ and $BC_{ff}$ were summed up and labeled as EC in the following Figures. PMF modelling was performed separately for each site. Statistical significance was validated with bootstrap higher than 80 % for each factor. Detailed methodology and results about these studies are given in their respective papers (Chevrier, 2017; Favez et al., 2016; Salameh et al., in prep.). It should be noted that glucose was not included in the final solution for any of these PMF, since it generally produced statistical instability of the solutions (this point is further discussed in Sect. 3.2).

The PMF analysis took advantage of the ME-2 algorithm to add constraints to different chemical profiles (see Tables SI-3 and SI-4 for details). Mainly soft constraints were applied in order to add some prior knowledge about the emission sources and "clean" the different profiles without forcing the model toward an explicit solution. In particular, the polyol concentrations were "pulled up maximally", while levoglucosan and mannosan were set to zero, and EC was "pulled down maximally" in the PBOA factor. This was achieved to avoid mixing with the biomass burning factor as well as possible influences of unrealistic high contributions of EC to PBOA. Other constraints were added parsimoniously to other factors, targeting specific proxies of sources (Table SI-4).

As for the general results of this large PMF study, we identified some well-known sources for almost all the sites (biomass-burning, road traffic, secondary inorganics, dust and sea salt). Two other less-common factors were identified for all sites: secondary biogenic aerosols (probably from marine origin), traced mainly by the presence of MSA, and PBOA, traced by the presence of more than 90% of the polyols total mass in the factor. Table SI-5 and Fig. SI-4 present more detailed description of the chemical tracers in each factor, together with their yearly average contribution for each site, respectively. Hereafter, only the PBOA chemical profile will be extensively investigated. The uncertainties of this PBOA factor are discussed below and its stability is presented in Fig. SI-5. Bootstrap analysis based on 100 resampling runs evidenced the very high stability of this PBOA factor since the PBOA initial constrained factor was mapped to PBOA bootstrap factor (BF) more than 99% of the time.

## 3. Results and discussion

### 3.1 Relative distribution between sugar alcohols and glucose

Figure 2 presents an overview of the relative mass concentration distributions of individual chemical species quantified at two sites with very different characteristics, an urban site in Grenoble and the rural site of OPE-ANDRA. Data are presented for the warmer season (e.g., during summer and fall), when concentrations were at their maximum (see Sect. 3.4). Glucose is the most abundant species measured (average $37.6\pm26.4$ ng m$^{-3}$), accounting on average for 25 % of primary sugar compounds (SC) total mass at both sites. Mannitol

(37.3±24.6 ng m$^{-3}$) and arabitol (32.0±22.2 ng m$^{-3}$,) are the second and third most abundant species, accounting
respectively for 25 and 23 % of SC mass. Trehalose is relatively abundant in samples from these two sites
(20.1±16.2 ng m$^{-3}$), accounting for 14 % of SC mass, but in general its concentration is frequently below the limit
of quantification for samples from other sites in France. The other identified polyols (i.e., erythritol, inositol,
glycerol, sorbitol, and xylitol) present lower concentration levels (4.9±2.1 ng m$^{-3}$), corresponding altogether to
13 % of SC total mass.
Such ambient mass concentration distribution patterns are similar (but with variable intensities) to those previously
reported for aerosol samples collected at various locations around the world. For example, Verma et al. (2018)
found that glucose, and arabitol together with mannitol, contributed to 16.7 and 48.1 %, respectively, of total
primary sugar compounds in aerosols from Chichijima Island. Similarly, Yttri et al. (2007b) showed that glucose
and the pair arabitol-mannitol were the main contributors of total primary monosaccharides and sugar alcohols in
aerosols collected from four various background sites in Norway. In addition, Carvalho et al. (2003) reported that
arabitol, mannitol and glucose are the most dominant primary sugar compounds in aerosols from rural background
and boreal forest sites in Germany and Finland, respectively.

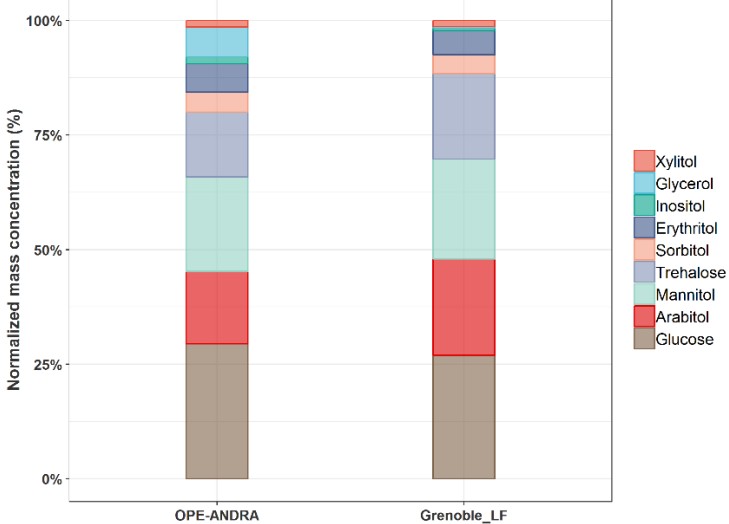


**Figure 2: Overview of relative mass distributions of individual primary sugar alcohols and saccharide compounds quantified in PM$_{10}$ samples at two sites over summer and autumn periods (June to November) corresponding to maximal atmospheric concentrations of sugar alcohols/saccharide compounds.**


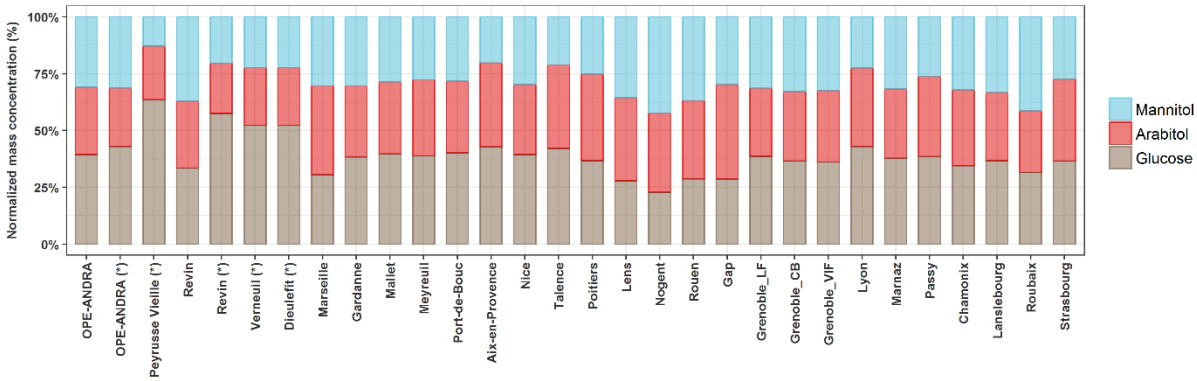


**Figure 3: Average mass concentration distributions of arabitol, mannitol, and glucose. Symbol (*) indicates PM$_{2.5}$ aerosol samples. Only data for warmer season (June to November), corresponding to maximal atmospheric concentration of polyols and glucose are shown.**

Although various primary sugar alcohols and saccharides have been detected and quantified for most of the
investigated sites, the following study focuses only on the three major and ubiquitous species, namely arabitol,
mannitol and glucose. Figure 3 presents their average relative contributions at all sites, for the warmer period,
displaying very similar features at a first glance. However, discrepancies could be observed from site to site, as
discussed in following sections.
**3.2 Relationships between selected primary sugar compounds**
Figure 4 summarizes linear correlations obtained between arabitol and mannitol concentrations at each site during
the warmer period. Medium to very high coefficients of determination could be observed ($0.58 \leq R^2 \leq 0.93$; $30 \leq$
$n \leq 143$ or $45 \leq n \leq 341$ for $PM_{2.5}$ and $PM_{10}$ series, respectively), with slopes in a rather narrow range (between
0.59 and 1.10), and quite low intercepts (always below 9 ng m$^{-3}$). Such covariations indicate that both species are
most probably co-emitted, by one or several type(s) of sources, at each site during the summer-autumn period.
These observations are in agreement with previous studies also showing strong covariations between arabitol and
mannitol (Kang et al., 2018; Verma et al., 2018; Zhu et al., 2015). Therefore, it seemed reasonable to consider both
species together, so that their concentrations are summed up and labelled "polyols" in the following sections.
Conversely, linear correlations between glucose and polyols concentrations are generally weaker
($0.10 < R^2 \leq 0.78$), with slopes varying over a much larger range (between 0.12 and 0.94), and variable intercepts
(between -5.6 and 16.4 ng m$^{-3}$). This suggests that glucose concentrations might follow a different pattern
compared to that of polyols, either due to different emission sources, or different chemical stability in the
atmosphere. It is therefore reasonable to keep glucose as a separate chemical species in the following discussion.
It should be emphasized that the variability in the slope of the regressions between the chemical concentrations is
most probably related to the emissions and atmospheric processing. Particularly in the case of mannitol and
arabitol, they may be influenced by biogenic or biotope characteristics. Nevertheless, no evident relationship
between the slope values and the typology or the geographical location of the sites could be observed (Figure 4).

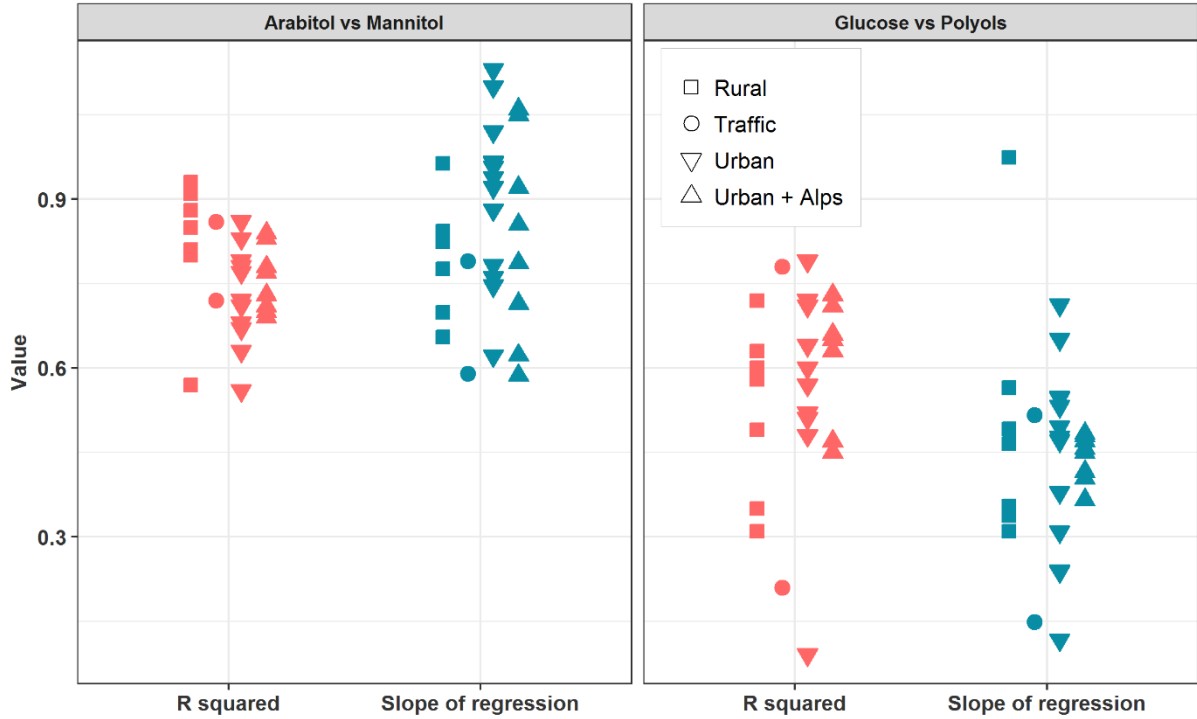


**Figure 4: Linear regression analysis between selected primary sugar compounds mass concentrations (i.e., arabitol, mannitol and glucose) during summer and autumn seasons (June to November), for all the sites considered in this study.**

### 3.3 Relative distributions between PM$_{10}$ and PM$_{2.5}$

Figure 5 shows the average PM$_{10}$ and PM$_{2.5}$ concentrations of polyols and glucose at OPE-ANDRA and Revin during the summer and autumn seasons. The polyols mass concentrations ranged from 7.5±10.9 to 27.8±33.3 ng m$^{-3}$ in PM$_{2.5}$, and from 48.9±38.2 to 73.5±61.8 ng m$^{-3}$ in PM$_{10}$, in Revin and OPE-ANDRA sites, respectively. PM$_{10}$-to-PM$_{2.5}$ ratios were then on average of about 3 to 5. Similar size distribution patterns, with variable intensity, were observed for glucose (Fig. 5). These results indicate that polyols and glucose are mainly associated with the coarse PM fraction. This observation is in good agreement with several previous investigations where polyols (especially arabitol and mannitol), together with glucose, were prevalent in the coarse fraction (Fu et al., 2012; Fuzzi et al., 2007; Pio et al., 2008; Yttri et al., 2007b). However, Carvalho et al. (2003) reported different size distributions for polyols and glucose, with variable fine or coarse mode maxima depending upon sampling location. For instance, maximum atmospheric concentrations of mannitol were associated to fine and coarse aerosols from boreal forest (Finland) and rural background sites (Germany), respectively. The authors hypothesized that these observations are due to different assemblages of dominant fungal biota (with variable aerodynamic characteristics) at different sites. Some other previous studies showed aerodynamic diameters typically ranging from 2 to 10 µm, even though a few airborne bacterial/fungal spores could exceed that size (Bauer et al., 2008a; Elbert et al., 2007; Huffman et al., 2012; Zhang et al., 2015).

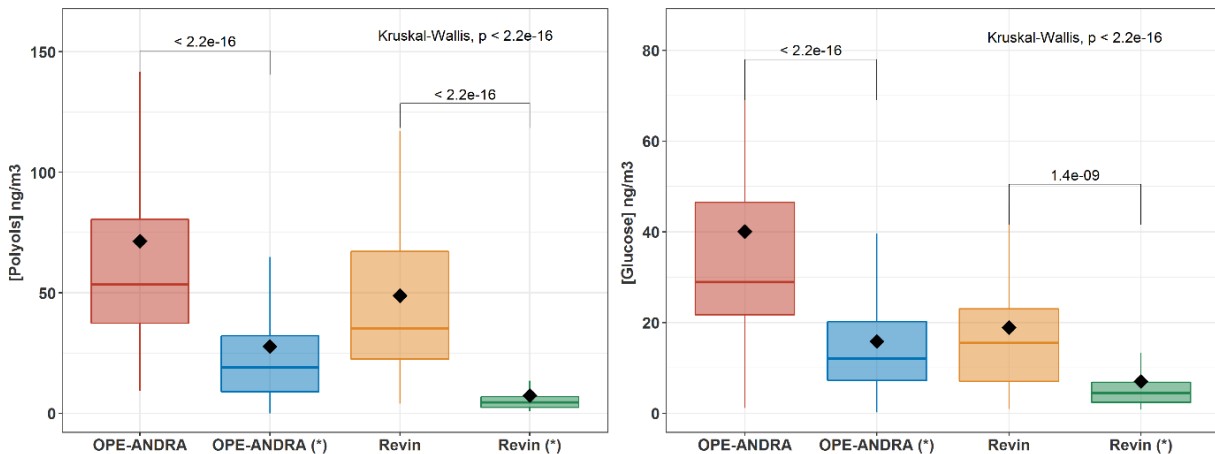


**Figure 5: Box plots of mass concentrations of polyols (left) and glucose (right) in PM$_{10}$ and PM$_{2.5}$ (with symbol (*)**
**samples). Black markers inside each boxplot indicate the mean concentration value, while the top, middle and bottom**
**lines of the box represent the 75$^{th}$, median and 25$^{th}$ percentile, respectively. The whiskers at the top and bottom of the**
**box extend from the 95$^{th}$ to the 5$^{th}$ percentile. Number of samples were N = 123 for OPE-ANDRA and N = 87 for Revin,**
**respectively. Statistical differences between average mass concentrations were analyzed with the Kruskall-Wallis**
**methods (p < 0.05).**
Hence, although if the precise mechanisms of atmospheric emission of particulate polyols and glucose are not fully
resolved, our observations are in good agreement with ambient mass concentrations of polyols and glucose being
likely associated with biological particles, as already suggested elsewhere (Fu et al., 2012; Verma et al., 2018;
Zhang et al., 2015). These species could enter into the atmosphere through either natural or anthropogenic
resuspension of surface soils and associated bacterial/fungal spores (containing polyols and primary sugar
compounds), or via a direct input resulting from microbial activities (e.g., sporulation). Another hypothesis would
be the abrasion of leaves and the subsequent release of microbial organisms and plant debris (Fu et al., 2012;
Medeiros et al., 2006; Simoneit et al., 2004b).
**3.4 Spatial and seasonal distribution of atmospheric concentrations**
**3.4.1 Spatial and seasonal patterns of polyol concentrations**
As illustrated in Fig. 6, significant concentrations of polyols were measured at each investigated site, evidencing
the ubiquity of these organic compounds. The annual average concentration levels of polyols measured in PM$_{10}$
aerosols at all sites (33.2±33.5 ng m$^{-3}$; see Table SI-2) are within the range previously reported for urban and rural
sites across Europe (Burshtein et al., 2011; Di Filippo et al., 2013; Pietrogrande et al., 2014; Yttri et al., 2007b,
2011). Additionally, polyols mass concentrations clearly exhibit seasonal trends, with variable intensity according
to the sampling sites. On a seasonal average, polyols are more abundant in summer (46.8±43.6 ng m$^{-3}$) and autumn
(43.0±36.7 ng m$^{-3}$), followed by spring (19.0±13.6 ng m$^{-3}$) and winter (16.2±11.5 ng m$^{-3}$). The average
concentrations of polyols are at least 2 to 3 times higher during summer or autumn months than during the cold
months, with a ratio that can be as high as 8 to 10.
Previous studies also reported similar seasonal variation pattern for urban and rural aerosol samples collected at
various locations. For example, Pashynska et al. (2002) measured higher atmospheric polyol (arabitol, mannitol)
contents during late summer and autumn, in Belgium. Several other studies reported higher concentrations of
polyols in summer than spring and winter time, in aerosols collected from Texas, USA and Jeju Island, respectively

(Fu et al., 2012; Jia et al., 2010a, 2010b). More recently, Liang et al. (2016) and Verma et al. (2018) also reported similar seasonal distributions for aerosols sampled in Beijing, China and north-western Pacific, respectively.

The higher atmospheric polyols concentrations observed are likely due to the increased contribution from metabolically active microbial derived sources (fungi, bacteria, green algal lichens) as a result of external stressors such as heat, drought and relative moisture. Indeed, fungal and prokaryotic cells activities, including their growth and sporulation, are promoted by high temperature and humid conditions occurring in summer and autumn (China et al., 2016; Elbert et al., 2007b; Jones and Harrison, 2004; Rathnayake et al., 2017).

As also evidenced from Fig. 6, atmospheric polyols concentrations do not present any significant seasonal differences related to the site typology (rural, traffic, urban sites with/without Alpine influences), or latitude. There is some tendency toward higher concentrations in summer in Alpine environments, but some other sites (like the rural site of OPE-ANDRA, in the North-East of France) can reach the same levels of concentrations. We tested several types of hierarchical classifications, including variables like monthly or seasonal mean polyols concentrations, the ratio arabitol-to-mannitol, or linear regression parameters (slope, R square) but none of them led to a simple clustering of the sites that would explain the variability of the concentrations.

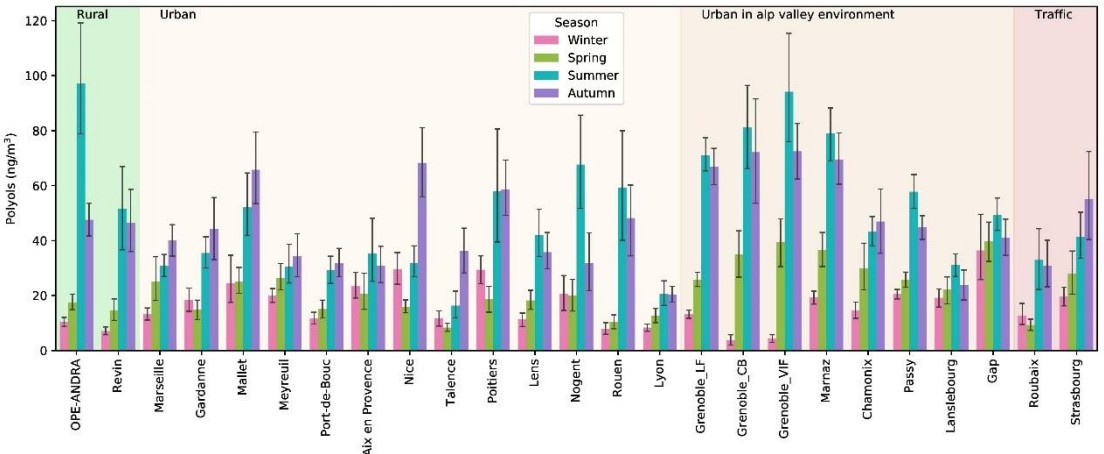

**Figure 6: Spatial and seasonal distributions of atmospheric polyol average concentrations (ng m$^{-3}$) for various types of sites in France. Error bars correspond to standard deviations calculated with seasonal concentrations. Years of PM sampling campaigns are not concurrent at all sites (see Fig. SI-1). The seasons were defined as follows: Winter = December to February, Spring = March to May, Summer = June to August, and Autumn = September to November.**

**3.4.2 Spatial and seasonal patterns of glucose concentrations**

The annual average concentrations of glucose measured in PM$_{10}$ aerosols at all sites (20.4±15.6 ng m$^{-3}$; see Table SI-2) are comparable to those previously reported for various sites across Europe (Alves et al., 2006; Theodosi et al., 2018; Yttri et al., 2007b, 2011). Likewise polyols, the atmospheric concentrations of glucose also display seasonal and site-to-site variations (Fig. 7). The ambient seasonal mean concentrations (with standard deviations) of glucose are maximum in summer (25.0±24.2 ng m$^{-3}$) and autumn (24.6±19.8 ng m$^{-3}$), followed by spring (15.8±12.4 ng m$^{-3}$) and winter (12.6±10.2 ng m$^{-3}$). The summer / winter ratio for glucose seems generally lower than that of polyols, with higher ratios in the Alpine areas than in other parts of France. However, as for polyols, it remains difficult to classify the sites according to any criteria linked to site typology or latitude.

The seasonal trend of glucose concentrations in the present work is similar to that recently observed for aerosols (PM$_{10}$ or total suspended particles) collected at various environmental background (suburban, urban and coastal)

sites around the world (Liang et al., 2016; Srithawirat and Brimblecombe, 2015; Verma et al., 2018). On average, a wide range of daily glucose concentrations (expressed as min-max, mean) in $PM_{10}$ (0.1-297.2 ng m$^{-3}$, 20.4±15.6 ng m$^{-3}$) were observed in the present study. These values are comparable to those in $PM_{10}$ (8.4-93.0, 47.0 ng m$^{-3}$) reported from an urban site in Norway (Yttri et al., 2007b). More recently, Liang et al. (2016) also reported similar concentrations in $PM_{10}$ (3.1-343.6, 46.2±27.5 ng m$^{-3}$) from Beijing (China).

The sources and formation processes of glucose in the atmosphere are not currently well known and are rarely discussed. Glucose is an important carbon source for soil metabolic active microbiota, and it is commonly present in vascular plants. Additionally, cellulose (a linear polymer made of glucose subunits linked by β-1,4 bonds) is one of the most important form of organic compounds in terrestrial ecosystems and a major plant structural polymer (Boex-Fontvieille et al., 2014). It can also be quite abundant in the atmosphere (Puxbaum and Tenze-Kunit, 2003). Hence, it is hypothesized that ambient glucose could be formed through active microbial (i.e., bacteria, fungi, etc.) enzymatic hydrolysis of cellulose in plant debris. Consistent with these observations, glucose could be released into the atmosphere from both vascular plant materials (e.g., leaves, fruits, pollens, etc.) growing in spring and decomposing in autumn/summer, and soil microbiota, as already suggested elsewhere (Di Filippo et al., 2013; Jia et al., 2010a; Medeiros et al., 2006; Verma et al., 2018; Zhu et al., 2015).

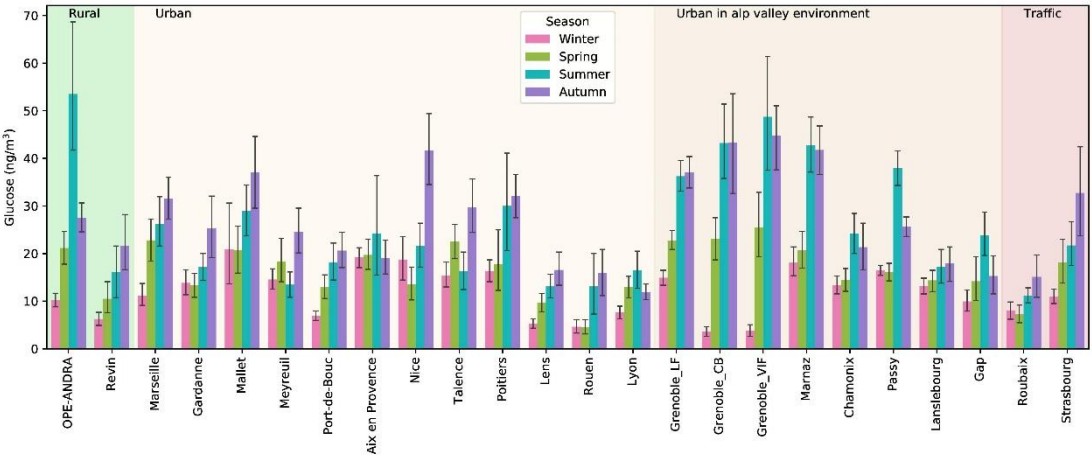

**Figure 7: Spatial and seasonal distributions of atmospheric glucose levels (ng m$^{-3}$) for various types of sites in France (except the site of Nogent, which presented too many missing values. Error bars correspond to standard deviations calculated with seasonal concentrations.**

### 3.4.3 Relative contributions to aerosol organic matter concentrations

The average contribution of polyols to the OM content of PM clearly displayed a seasonal behavior, as shown in Fig. 8. Here again, contributions are 2 to 10 times higher during summer and autumn compared to winter and spring, consistent with the assumption of higher emissions during these periods. The seasonal mean contribution of polyols to OM fluctuates from site to site, and accounts for 0.1 to 2.1 % of overall OM for these French sites (Fig. 8). Similarly, the seasonal mean concentrations of polyols together with glucose represent between 0.2 to 3.1 % of total OM at these sites (Fig. SI-2). However, on a daily basis (Samake et al., in prep.), atmospheric polyols mass concentrations can represent up to 6.3 % of total OM in $PM_{10}$, indicating that polyols can be amongst the major molecular species identified in aerosol organic matter (Fig. SI-3). Again, we could not find any simple way to group the sites according to their characteristics (typology or latitude, or climatic region), in order to better

understand the drivers behind the variability of this mass fraction. Further studies are currently conducted using multi criterion examinations.

The seasonal polyols-to-OM distribution patterns in this study are comparable to those found for different urban or rural sites in Europe (around 0.2 to 2.5 % of OM) (Pashynska et al., 2002; Yttri et al., 2007b). Zhu et al. (2015) also reported similar seasonal polyols-to-OM contribution trend for aerosols sampled at Cape Hedo (coastal site, Japan).

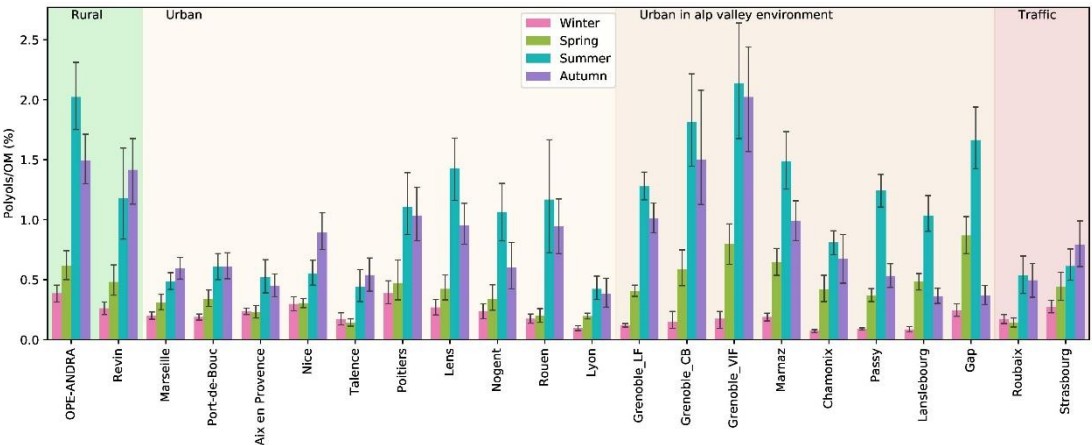

**Figure 8: Spatial and seasonal distributions of mean contributions (in %) of polyols to the organic matter content of PM for various types of sites in France. Daily time-series of organic carbon (OC) were not available for the following sites: Gardanne, Mallet, and Meyreuil. Error bars correspond to standard deviations calculated with seasonal concentrations.**

**3.5 Primary biogenic factor in PMF studies**

The sum of polyols (arabitol + mannitol) represents only a small fraction of the total OM. However, as proxies of PBOA, they are most probably emitted with other chemical species. Emission from biological particles is a complex topic since it may include a wide variety of compounds, both organic and inorganic (Elbert et al., 2007; Zhang et al., 2015). Moreover, it is not clear if polyols are mainly emitted directly in the atmosphere or are linked to other materials, for example with soil dust during resuspension processes. To investigate the relationship between polyols and other molecular tracers of emission sources, it would have been possible to perform simple correlation analysis with individual chemical species. This approach has the disadvantage of being a one-to-one relation and thus highly sensitive to the dynamics of all PM emission sources, not only the one we are interested in. Alternatively, another way is to use a PMF approach, also based on correlations but including much more information on the temporal variations of the different sources influencing the PM chemistry at a given receptor site.

As mentioned in Sect. 2.3, the PMF results used in this study include sites of different typologies (rural, traffic, urban sites in Alpine valley environments, etc.) for 16 different locations spread over France and part of the current dataset. At each site, the PMF studies allowed to identify a PBOA factor, characterized by the presence of more than 90 % of the total polyols content (sum of arabitol, mannitol and sorbitol), as presented in table SI-5 and Fig. SI-6. Moreover, the sensitivity of this factor to random noise in the data was investigated thanks to randomly re-sampling the input matrix of observation. In PMF analysis, this is done via the bootstrap method (Paatero et al., 2014) in the constrained run. The PBOA factor was always mapped to itself for 13 of the sites and quasi-always (97%) for the last three ones. It means that the PBOA factor does have a very high statistical stability since it never

swaps with another factor (see Fig. SI-5). Hence, the chemical composition of this factor may be informative to
investigate the PBOA source components (Table SI-6), and to evaluate the importance of PBOA emissions in
terms of OM mass apportionment.

### 431     3.5.1. Contributions of PBOA to OM and polyols to PBOA

Altogether, the results from the16 sites highlight the importance of the PBOA source contribution to total OM. As
shown in Fig. 9, the OM apportioned by the PBOA factor represents a significant fraction of the total OM mass
on a yearly average (range 6–28 %; average 13±6 %). When considering only the summer period (June-July-
August), this contribution is even larger and can exceed 40 % of the total OM at sites in the Alpine area (Marnaz,
Passy, Chamonix, Grenoble_LF) which are partially protected from large regional influences due to the local
topography. This result may be nuanced, in particular during summer, since some extent of mixing between PBOA
and Biogenic Secondary Organic Aerosols (BSOA) cannot be entirely excluded. However, several evidences tend
in favor of a non-significant mixing between BSOA and PBOA. First, the ratio of polyols-to-$OC_{PBOA}$ shows a low
variability from site to site, while it is unlikely that such a secondary process led to the same amount of OC for all
sites since they present different meteorology, sunshine duration, etc. Second, the bootstrap analysis does not show
any "swap" between factors for the PBOA profile for all sites, indicative of a well-defined factor (see Fig. SI-5).
Finally, the $OC_{PBOA}$-to-polyols ratio in this work (about 16) is in the range of ratio expected for fungal spores (12
-27, when arabitol and mannitol are considered together) (Bauer et al., 2008a; Yttri et al., 2011).
Interestingly, some previous work using the same samples from the sites in the Arve valley (Passy, Chamonix)
showed that about 90 % of the OM is from modern origin (using [14]C measurements) during summer, with no
apparent correlation between this modern carbon and polyols concentrations (Bonvalot et al., 2016). Hence, despite
being an important contributing source, PBOA is not the major biogenic source in this type of environment.
Interestingly, opposite to the case of the Alpine valleys where this proportion is the highest, the ratios $OM_{PBOA}$-to-
$OM_{total}$ are amongst the lowest for coastal environments (Talence, Marseille, Nice), a possible indication that the
marine environment is not a large emitter for these species. Recently, much lower concentrations of polyols in
aerosols from marine environments than those in terrestrially influenced sites were also reported off the coast of
Japan, also suggesting a higher contribution from terrestrial sources (Kang et al., 2018).
As illustrated in Fig. 9, polyols represent only a small fraction of the OM apportioned in the PBOA factor (1.2 %–
6.0 %; average 3.0±1.5 %) for the 16 studied sites. This variability is indeed rather small, considering the wide
range of sites and the diversity of other potential sources (on average 8 to 10 PMF factors were obtained for the
different sites). Indeed, this narrow range of the polyols fraction to the $OM_{PBOA}$ highlights the stability of the
chemical profile of this source over a large regional scale. It indicates also that, if polyols are good proxies of the
PBOA sources, a large amount of other organic species are co-emitted, that still remain unknown.

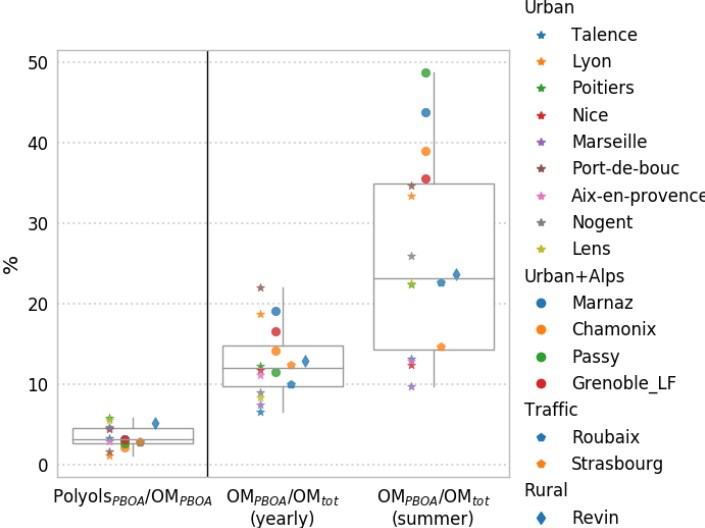


**Figure 9: Mass contribution of polyols to OM in the PBOA factor, and relative contributions of the OM_PBOA factor to the total OM in PM for the 16 studied sites where PMF model was run, over the year and summertime only. Stars and circle refer to urban sites without/and with Alpine valley influence, respectively. Pentagon corresponds to traffic sites and diamond to rural sites.**

**3.5.2 PBOA profile constituents and emission process**
Figure 10 shows the contribution (in µg of species per µg of PM in the PBOA factor profile) of each chemical
species included in the averaged PBOA factor from the 16 PMF studies. The principal contributors are OC and
EC, and significant fractions of crustal material also appear ($Na^+$, $K^+$, $Ca^{2+}$, Al, Ba, Cu, Fe, Mn, Ti, Zn) as well as
secondary elements such as nitrate and sulfate. However, EC appears to be highly variable both within and between
sites under consideration. The reader may refer to figure SI-7 for an estimation of the EC mass uncertainties in the
different sites.  On average, the PBOA factor does not comprise a large fraction of metals and trace elements, most
of them being below 1 pg µg$^{-1}$. Here again, the low variability of the PBOA chemical profile encountered across
a large array of sites is remarkable.

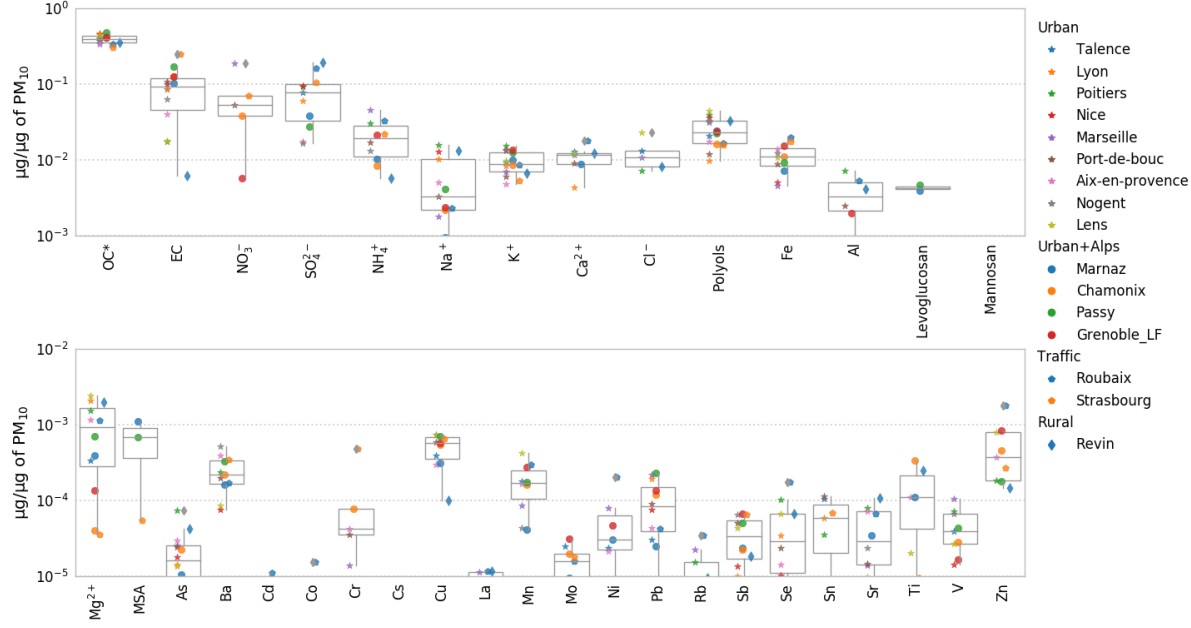


**Figure 10: PMF chemical profile of the PBOA factor in the DECOMBIO and SOURCES programs expressed as a fraction of the PM mass. Values lower than a few pg µg$^{-1}$ are not displayed on purpose. For each boxplot, the top, middle**

**and bottom lines represent the 75ᵗʰ, median and 25ᵗʰ percentile, respectively. The whiskers at the top and bottom of the**
**box extend from the maximum to the minimum. OC\* corresponds to the bulk organic carbon fraction minus the carbon**
**in the characterized organic species.**

The contribution from some crustal material could agree with the coarse mode distributions of polyols (Sect. 3.3)
and could be indicative of an emission process with the entrainment of spores with soil dust resuspension. To
investigate the importance of mineral dust in the PBOA factor, we clustered the chemical components of PM from
PBOA into 7 classes: OM (= $1.8 \times$ OC), EC, $NO_3^-$, $NH_4^+$, non-sea-salt sulfate (nss-$SO_4$), sea-salt, and dust. nss-
$SO_4^{2-}$ is calculated from the measured $SO_4^{2-}$ minus the sea-salt fraction of $SO_4^{2-}$ ($nssSO_4^{2-} = SO_4^{2-} - ssSO_4^{2-}$
where $ssSO_4^{2-} = 0.252 \times Na^+$) according to Seinfeld and Pandis (1997). The sea-salt fraction is calculated
according to Putaud et al. (2010): $sea - salt = Cl^- + 1.47 \times Na^+$. Finally the dust fraction is estimated thanks
to Putaud et al. (2004b) as: $dust = (nss - Ca^{2+}) \times 5.6$ with nss-$Ca^{2+}$ stands for non-sea salt $Ca^{2+}$ and is
computed thanks to $nss - Ca^{2+} = Ca^{2+} - Na^+/26$. We note that the conversion coefficient provided by Putaud
et al. (2004b) may be influenced by an extreme value and then gives only a low estimate of dust resuspension.
Figure 11 presents the normalized average contributions of these 7 classes to the PBOA mass for the 16 sites with
PMF modelling. It clearly reveals that the PBOA factor is dominated by contributions from OM (78±9 %),
followed by EC (9±7 %), and only a minor contribution from the dust class (3±4 %).
The large value for the contribution of EC is driven by two high values obtained at the sites of Strasbourg (that
reaches 25%) and Chamonix (18%) both influenced by direct and indirect traffic emissions. However, 6 other sites
present no EC in PBOA. Moreover, the uncertainties of EC in the PBOA profile of Strasbourg and Chamonix is
rather high (between 5 to 30% of PM mass at Strasbourg, see SI-7). On a yearly average, EC apportioned by this
factor (0 to 400 ng m⁻³ depending on the site) is close to the rural EC background in France of about 300 ng m⁻³
(Golly et al., 2018).
This result on the general chemical profile of the PBOA factor, with a low crustal fraction, tends to infirm the
hypothesis of an emission process associating PBOA material with mineral dust resuspension. Indeed, our findings
rather suggest that a main part of PBOA (and polyols) is most likely associated with biological particle direct
emissions. It leaves only a minor fraction that could be linked to the mechanical resuspension of PBOA with crustal
elements. Some minor fraction of EC in this factor could come from resuspended EC-containing dust particles
being accumulated in topsoil as demonstrated in previous works (Forbes et al., 2006; Hammes et al., 2007; Zhan
et al., 2016). Hence, the origin of the larger fraction of the contribution of EC remains unknown. Our conclusions
are in good agreement with those made by Jia and Fraser (2011), based on the concentrations of these chemicals
in different types of samples: i.e., size-fractionated (equivalent to $PM_{2.5}$ and $PM_{10}$) soil, plant, fungi, atmospheric
$PM_{2.5}$ and $PM_{10}$. They found that the ambient concentrations of primary saccharide compounds at the suburban
site of Higley (USA) are typically dominated by contributions of biological materials rather than resuspension of
soil dust particles and associated microbiota.

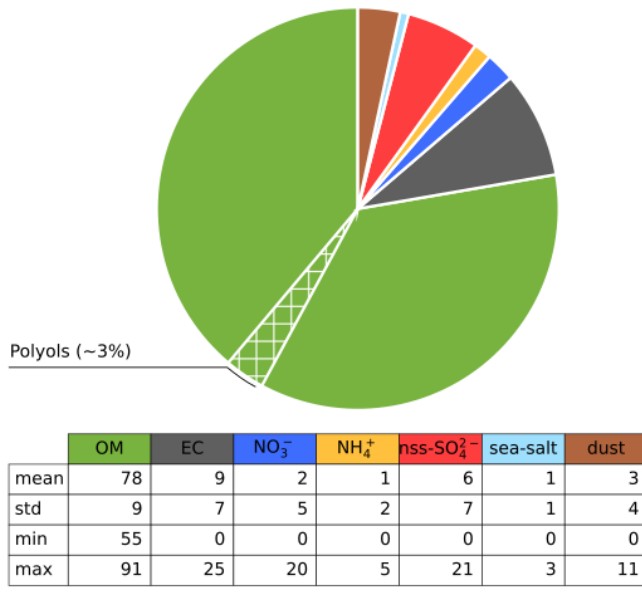

| | OM | EC | $NO_3^-$ | $NH_4^+$ | nss-$SO_4^{2-}$ | sea-salt | dust |
|------|-----|----|----|----|----|----|----|
| mean | 78 | 9 | 2 | 1 | 6 | 1 | 3 |
| std | 9 | 7 | 5 | 2 | 7 | 1 | 4 |
| min | 55 | 0 | 0 | 0 | 0 | 0 | 0 |
| max | 91 | 25 | 20 | 5 | 21 | 3 | 11 |

**Figure 11: Average contribution (%) of species in the PBOA factor for the sites in SOURCES and DECOMBIO. The hatched area represents the proportion of the OM apportioned by the polyols (see text for reconstruction method).**

## 4. Conclusion

The contribution of primary biogenic organic aerosols to PM is barely documented in the scientific literature. The present study aimed at providing a large overview of the spatial and temporal evolution of concentrations and contributions to aerosol organic matter of dominant primary sugar alcohols and saccharide compounds, for a large selection of environmental conditions in France. With 28 sites and more than 5,340 samples, it is to our knowledge the most comprehensive dataset for these compounds. The main results obtained indicate that:

- among the identified polyols, arabitol together with mannitol are the major species by mass, with lesser amounts of others polyols (e.g., erythritol, inositol, glycerol, sorbitol, and xylitol). Glucose is the dominant primary monosaccharide and its relative abundance is comparable to the sum of arabitol and mannitol;

- the two main polyols (arabitol and mannitol) together with glucose are mainly present within the coarse aerosol mode;

- at nearly all sites, ambient levels of the main polyols and glucose displayed clear seasonal variation cycles, with a gradual increase from spring and maximum in summer and autumn aerosols, followed by a sudden decrease in late autumn, and a winter minimum;

- atmospheric concentrations of the main polyols and glucose fluctuate according to site and season, and account each for between 0.1 to 2.1 % of OM on a seasonal average basis at these French sites;

- ambient mass concentrations of arabitol and mannitol are comparable. Meanwhile, they display very good temporal covariation, with ratios varying between sites. Conversely, linear correlations between the main polyols and glucose concentrations are much lower, suggesting different atmospheric sources, or atmospheric processes;

- arabitol and mannitol are efficient organic markers for PBOA. PMF studies of the yearly series from 16 sites give contributions of the primary biogenic emission (traced with the main polyols) to the total OM around 13±6 % on a yearly average and 26±12 % during summer, thereby showing that PBOA is an important source of total OM in $PM_{10}$ for all sites across France. Furthermore, the average PBOA chemical source profile is made out of a very large fraction of OM (78±9 % of the total PBOA mass on average), suggesting it is mainly related to direct biogenic emissions from biological particles. Noteworthy, the presence of BSOA within the PBOA factor, particularly during summer could not be fully ruled out and further works using additional organic tracers (such as 3-methylbutanecarboxylic acid, pinic acid, and/or cellulose) are still needed to solve this issue. Additionally, the low crustal fraction indicates that this factor is weakly linked to soil dust resuspension associated with biological material;

- however, the PBOA source remains chemically poorly characterized as the main polyols represent only a small fraction of its total OM mass (3.0±1.5 % on average);

- despite comparable high concentrations in the atmosphere, the sources and processes leading to glucose concentrations and seasonal evolutions are still elusive. Indeed, the different PMF performed with glucose in input variable do not lead to a statistically stable solution;

Further investigations of the emission pathways and chemical characterization of the PBOA source associated with polyols are on-going, which may improve our understanding of its dynamic at various geographical scales, for a potential implementation in emission models in the future.

**Acknowledgements:** The PhD of AS and SW are funded by the Government of Mali and ENS Paris, respectively. We gratefully acknowledge the LEFE-CHAT and EC2CO programs of the CNRS for financial supports of the CAREMBIOS multidisciplinary project. Samples were collected and analyzed in the frame of many different programs funded by ADEME, Primequal, the French Ministry of Environment, the program CARA ledby the French Reference Laboratory for Air Quality Monitoring (LCSQA), and actions funded by many AASQA, ANDRA, IMT Lille Douai (especially Labex CaPPA ANR-11-LABX-0005-01 and CPER CLIMIBIO projects), etc. Analytical aspects were supported at IGE by the Air-O-Sol platform within Labex OSUG@2020 (ANR10 LABX56). We acknowledge the work of many engineers in the lab at IGE for the analyses (A. Wack, C. Charlet, F. Donaz, F. Masson, S. Ngo, V. Lucaire, and A. Vella), as well as B. Malet and L. Y. Alleman (IMT Lille Douai) for analyzing trace and major elements in aerosols from the northern sites. Finally, the authors would like to kindly thank the dedicated efforts of many other people at the sampling sites and in the laboratories for collecting and analyzing the samples. The authors would like to thank the editor and several anonymous referees for comments that greatly improved the manuscript.

**Author contributions:** JLJ was the supervisor for the PhD for AS, FC, SW, and for the post-doc of DS. He directed all the personnel who performed the analysis at IGE. He was coordinator or principal investigator (PI) of the programs that generated the data for 18 of the 28 sites in this study (OPE-ANDRA, Part'Aera, CAMERA, SRN 2013, 3 Villes PACA, DECOMBIO, QAMECS) and co-PI for programs for 5 other sites. He is the coordinator for the CNRS LEFE-EC2CO CAREMBIOS program that is funding the work of AS. GU was the co-supervisor for the PhD of AS and SW. OF is the coordinator of the CARA program, (co-)funding and supervising the filter sampling and chemical analyses at 12 of the 28 sites. EP, OF, and VR supervised the PhD of DMO who

investigated the 5 sites in northern France. Finally, JLB was the coordinator (program Lanslebourg) or partner of
several programs whose data were used in this study (OPE-ANDRA, Part'Aera, 3 Villes PACA, DECOMBIO),
and OF was the coordinator of the SOURCES program, which includes the work of DS as a post-doctoral fellow
under the supervision of JLJ to gather and prepare most of the datasets used in the present studies.
All authors from the ANDRA (#5) and AASQA (#6 to 13) are representatives for each network that conducted the
sample collection and the general supervision of the sampling sites.
FC and DS ran the PMF analysis. AS, SW and JLJ processed the data and wrote up the manuscript. All authors
reviewed and commented on the manuscript.
**Competing interests:** The authors declare that they have no conflict of interest.

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
