# Peer review of "Polyols and glucose particulate species as tracers of primary 1 biogenic organic aerosols at 28 French sites 2"

_Atmospheric Chemistry and Physics, 2018_

## Referee Comment (RC1) · Anonymous Referee #4 · 29 Oct 2018

Comments on "Polyols and glucose particulate species as tracers of primary biogenic organic aerosols at 28 french sites" by Samake et al. This is a rich manuscript pooling together datasets from 28 sampling sites across France, focusing o polyols and glucose in the atmosphere. I find the manuscript to fit the scope of the journal, and to be generally well-written. I do have some fairly significant concerns about the analysis and technical comments which should be addressed prior its publication.

General comments:

1. I find the title misleading, as there is not really use of polyols and glucose as tracers of PBOA. To achieve that, both had to be quantified and recommendations provided on

how the formers can be used to estimate the latter. Instead, the manuscript is rather descriptive on polyols and glucose, and results from a largely unclear PMF analysis is given, which leads to my next comment.

2. The PMF analysis is overwhelmingly under-reported and under-explained, given that only its main results are presented. The analysis is actually referred to a report on a url which is no longer valid, or refers to publications in preparation, which is not acceptable, given that one cannot first publish the results and later the analysis. If the authors decide to keep PMF analysis for the revised version of this study, further (and complete) details of the analysis must be provided.

3. Please present your results (e.g. Fig. 6) limited only to PM10 sampling, as it is bound to represent more closely the actual atmospheric concentration, instead of being limited by too low sampling cut-off for the species studied. I recommend maintaining though section 3.3 (PM25/PM10 comparison) to report fine vs coarse mode analysis.

4. I find the sampling site denominations used here unsuitable. Urban sites are typically strongly impacted by traffic emissions, so their distinction feels arbitrary. And why rural? Do you mean background? From those denominations it feels like it is lacking filter sampling at forested sites, for example. An improved description of the sampling sites is necessary to better understand its somewhat unexpected results.

5. There is certainly a lot to gain from combining several sampling sites, but I find that the analysis has become too shallow, unfortunately. Could you also focus on one sampling site and add more analysis (e.g. comparison with FBAP, total number, other species, wind direction/speed, backtrajectory, etc.) to try to better understand what is driving polyols and glucose atmospheric concentration? The manuscript seems to bring more questions than to answer at this point. Especially when it is kept fairly general (unclear PMF, unclear sampling periods, unclear site characterizations, etc.).

Specific/technical comments:

[Figure]

Abstract: Unclear why dust ressuspension would be linked to PBOA factor.

L.53: PM affects climate, not necessarily negatively.

L.57: please refer to a more recent reference for carbonaceous matter.

L.57-L.66 I suggest focusing on OM on the introduction, rather than OC, an artificial species from analytical limitations.

L.63: a significant fraction of OM can be associated with . . .

L.72: Please specify in which environments you are referring this figure, including atmospheric layer and aerosol sizes.

L.74-76: And fluorescent techniques?

L.79: Unclear how atmospheric transport complements sources and abundances.

L.101: Datasets

L.104: Please define atmospheric emission pathway. Do you mean the processes the plant underwent to emit polyols?

L.132: Please define "very rural".

L.152: Please state that this number typically ranges from 1.2 to 2, so the estimates here represent an upper value of OM, thus a lower estimate of the contribution of PBOA.

L.185: extra space before comma.

L.186. Define JRC

L.194: It is unclear why mix up filter-based BC with already quantified thermo-optical EC. Or there was no EC from DECOMBIO project? Please clarify.

L.200: See comment #2

L.211: Range values refer to min/max? In terms of readability I prefer you remove this info and present only avg±std.

L.212: Please define Primary Sugar Compound (SC).

L.228: Please increase axis font sizes.

L.233: The asterisk is hard to readily identify. Please show only PM10 cutoff filters on this figure.

L.233: The selected period feels somewhat arbitrary, thus lacking a clear definition of what is shown. Please be more direct on the chosen periods (dd-mm-yyyy) and criteria applied.

L.255: Please add the information of their estimated atmospheric lifetime.

L.256: It feels like a weak hypothesis to me, from the PBOA perspective, could it be emission ratios change with wind speed, temperature, RH? If focusing on comparable season/meteorology, could the correlation be improved, given distinct emission pathways? And how about interferences from other sources? Is it mixing PM2.5 samples?

L.267: To improve readability, please remove SD and describe only the four average values of both sampling sites, given the interest is the distribution of fine vs coarse mode.

L.290: Please remove "compartment".

L.282: Please indicate the number of samples used on this analysis.

L.301: Does it make sense that PBOA-related polyols are "only" 2-3 times higher in summer in comparison to winter time? The trend behind concentrations in "rural", "urban" or "traffic" feels inconsistent with PBOA interpretation.

L.404: In which time series?

L.440 Please correct sea-salt and not " sea minus salt".

L.445: Unclear sentence.

---

## Referee Comment (RC2) · Anonymous Referee #3 · 6 Nov 2018

General Comments.

This manuscript presents an interesting data set, treating aerosol PM10 and PM2.5 composition for a number of Traffic, Urban and Rural sites across France that is important for the understanding of aerosol sources over continental west European areas. From this data set the manuscript focus specially in the polyols and sugar components with the objective of determining the importance of this group of organics and their sources in the atmospheric aerosol loading. Unhappily the manuscript is not well presented. The first part, 3.1 to 3.4 sections, is mostly descriptive, showing average values and variability for polyols and glucose across sizes, seasons and regions. The

authors try to evidence the importance and contribution of those compounds to the aerosol loading in a somehow enthusiastic and forced way. They have the tendency to present more maximum concentrations than average values. The second part, section 3.5, deals more specifically with the evaluation of the Polyols source composition and contribution to the aerosol loading, using mainly PMF analysis. However the authors only show the PMF results concerning the source associated to polyols, remitting the reader to an unpublished manuscript for further scrutinizing of the aerosol total source apportionment and this is not acceptable. Furthermore, the so-called PMF calculated PBOA source factor has a mass that is more than 30 times higher than the measured polyols without a clear explanation about how can this result from primary particulate biomass emissions. I have some doubts about the correctness of this source factor as discussed further in the Specific Comments part. Therefore, I recommend that the manuscript is reorganized and modified in order to provide a more detailed information and discussion of the sources of the atmospheric aerosol and the contribution and importance of polyols and sugars as sources of the particulate pollution.

Specific Comments.

Line 174- "130 Different chemical species"? I only counted around 40.

Line 175- Table S2 instead of S1?

Line 194- "BC" instead of "EC"?

Line 200-204- The imposition of these constrains may not influence artificially the composition results? Anyway, the PBOA source calculated still has important contributions of unexpected EC.

Line 2017- Which is the necessity of having a Figure S3 that is very similar to Figure 2? Substitute Figure 2 in the text by Figure S3.

Line 232 Figure 3- If possible harmonize colors in this Figure with colors in Figure 2, for Mannitol, Arabitol and Glucose.

[Figure]

Line 233 Add "Average" initially to the sentence.

Line 256- Could you give some more information and reasoning about the removal of glucose from the PMF treatment?

Line 258- Change to "...the variability in the slope of the regressions between the chemical concentrations is most probably..."

Line 269- change to "to-PM2.5 ratios were..."

Line 356-366- There is a lack of information concerning average Polyols and glucose concentration values for the total sampling sites and perhaps either to each one of the four classified station types. A column to the right of the Figures 6 -8 with average values for the station ensemble would be informative.

Line 429 Figure 10- Which is the meaning of "*" in OC?

Line 447-449- The mass of EC contribution to the PBOA factor shown in Figure 11 is 3 times higher than the mass of soil estimated. Then, it is impossible to conclude anything about EC in soil from this data.

Line 453 Figure 11- The PBOA factor has an important contribution of EC (ratio of OC/EC equal to approximately 4.8, similar to values found in secondary organic aerosol formation). Therefore in my opinion this PBOA factor is probably highly contaminated with secondary organic material. That may explain the more than 30 times higher PBOA mass than the mass of polyols. However a more well based evaluation is impossible given the lack of complementary information from the PMF source apportionment.

---

## Author Comment (AC1) · 18 Dec 2018

We thank the reviewer for his/her attention to our manuscript. We reworked and re-arranged it in many places, in order to take into account all the general remarks and specific comments below, as well as those of the reviewer # 4. Particularly, we are now providing a more in-depth presentation of the overall PMF methodology and results. The detailed responses to the comments are given below, point by point, in blue color, including changes directly made to the manuscript, in red color. A list of references used to address Reviewer's comments is given at the end of the present response letter.

[Figure]

This manuscript presents an interesting data set, treating aerosol PM10 and PM2.5 composition for a number of Traffic, Urban and Rural sites across France that is important for the understanding of aerosol sources over continental west European areas. From this data set the manuscript focus specially in the polyols and sugar components with the objective of determining the importance of this group of organics and their sources in the atmospheric aerosol loading. Unhappily the manuscript is not well presented. The first part, 3.1 to 3.4 sections, is mostly descriptive, showing average values and variability for polyols and glucose across sizes, seasons and regions. The authors try to evidence the importance and contribution of those compounds to the aerosol loading in a somehow enthusiastic and forced way. They have the tendency to present more maximum concentrations than average values. The second part, section 3.5, deals more specifically with the evaluation of the Polyols source composition and contribution to the aerosol loading, using mainly PMF analysis. However the authors only show the PMF results concerning the source associated to polyols, remitting the reader to an unpublished manuscript for further scrutinizing of the aerosol total source apportionment and this is not acceptable. Furthermore, the so-called PMF calculated PBOA source factor has a mass that is more than 30 times higher than the measured polyols without a clear explanation about how can this result from primary particulate biomass emissions. I have some doubts about the correctness of this source factor as discussed further in the Specific Comments part. Therefore, I recommend that the manuscript is reorganized and modified in order to provide a more detailed information and discussion of the sources of the atmospheric aerosol and the contribution and importance of polyols and sugars as sources of the particulate pollution.

Specific comments:

Line 174- "130 Different chemical species"? I only counted around 40. The reviewer is right that this table only presented around 40 chemical species, since not all of the analyzed chemical species were included in the PMF runs. This is now explained in the manuscript: line 195, page 6.

[Figure]

Line 175- Table S2 instead of S1? This has been rephrased. It now corresponds to the Table SI-6.

Line 194- "BC" instead of "EC"? Filter analysis for EC-OC and aethalometer measurements were conducted simultaneously for the 3 DECOMBIO sites within the Arve valley (Chevrier, 2016). Aethalometers give the total BC, thus enabling the decomposition of BC concentrations into its two main constituents: wood-burning BC (BCwb) and fossil-fuel BC (BCff) (Sandradewi et al., 2008). Considering the very specific context of this mountainous valley (with a large influence of meteorology on atmospheric concentrations in winter), the PMF runs with better results in term sof statistical stability and geochemistry were the one with BCwb and BCff. However, for graphical simplicity, BCwb and BCff are summed up and labelled as EC in the present study. This point is further detailed in the response to the comment 14 of Anonymous Referee #4

Line 200-204- The imposition of these constrains may not influence artificially the composition results? Anyway, the PBOA source calculated still has important contributions of unexpected EC. The PMF runs take advantage of the ME2 algorithm to add constraints to different chemical profiles (see the tables SI-3 to SI-5 for in-depth details). Mainly soft constraints were applied in order to add some prior "expert knowledge" about the emission sources and their chemical profiles, and "clean" the different profiles without forcing the model toward an explicit solution. This is now classical in many PMF work (Bozzetti et al., 2017; Daellenbach et al., 2017; Srivastava et al., 2018a, 2018b; Weber et al., 2018). This point is also further detailed in the response to the comment 15 of Anonymous Referee #4. Also, details about the contribution and uncertainties of EC over the sampling sites have now been addressed in Fig. SI-7. As evidenced in Fig. SI-7, EC apportioned by PBOA appears to be variable both within and between the sites under consideration. Indeed, the average contribution of EC calculated for this profile is largely influenced by two values of about 25% for the traffic site at Strasbourg and 18% at Chamonix (2 sites influenced by direct and indirect traffic emissions), but 6 other sites present no EC in this factor (hence the large standard deviation 9±7 % mentioned in the main text, Fig.11). In term of mass (SI-7a), the PBOA factor is making up between 0 to 400 ng m-3 of EC on yearly average depending on the site, which is close to the background value of remote sites in rural France of about 300 ng m-3 (Golly et al., 2018). There are indeed some uncertainties on how the PMF method is redistributing the components of this background PM with no strong specific chemical signature among all the extracted factor. Further, one should keep in mind that the factors defined in a PMF are characteristic of a "source", a mixing of sources and/or atmospheric processes as seen at the receptor site. Hence, they may be somewhat influenced by processes taking place during atmospheric transport (i.e., aging, mixing, etc.) and may not present a chemical signature as pristine as that right at the emission point. All of these could justify the presence of EC in the chemical profiles.

Line 2017- Which is the necessity of having a Figure S3 that is very similar to Figure 2? Substitute Figure 2 in the text by Figure S3. The Fig. S3 has now been removed, as suggested by the reviewer.

Line 232 Figure 3- If possible harmonize colors in this Figure with colors in Figure 2, for Mannitol, Arabitol and Glucose. We agree with reviewer#3 and harmonized the colors between Fig.2 and Fig.3, for glucose, arabitol and mannitol.

Line 233 Add "Average" initially to the sentence. The term "average" has been added

Line 256- Could you give some more information and reasoning about the removal of glucose from the PMF treatment? We can note that the papers dealing with glucose in atmospheric PM are really rare and, to the best of our knowledge, there is currently no discussion or hypothesis on the origin or the stability of glucose in PM. As mentioned in the text, adding glucose in the PMF treatment always led to PMF solutions that were unstable or did not make sense in a geochemical way. Particularly, glucose showed no tendency to mix with the other polyols in a single factor. So far, we are not able to provide a proper explanation for this behavior, despite many investigations on the data base. It may be that the factor that would include the larger part of glucose (if

any) could present a really variable chemical profile across the year, while stability is a strong hypothesis underlying the PMF method. This would prevent getting a well identified factor.

Line 258- Change to ". . .the variability in the slope of the regressions between the chemical concentrations is most probably. . ." This information has been added.

Line 269- change to "to-PM2.5 ratios were. . ." This correction has been added.

Line 356-366- There is a lack of information concerning average Polyols and glucose concentration values for the total sampling sites and perhaps either to each one of the four classified station types. A column to the right of the Figures 6 -8 with average values for the station ensemble would be informative. The information concerning the annual average polyols and glucose concentrations for the total sampling sites, and for each one of the four classified station typologies is now provided in Table SI-2. This information is now discussed in the manuscript (lines 341-344, page 11; lines 373-375, page 12).

Line 429 Figure 10- Which is the meaning of "*" in OC? OC* corresponds to the bulk organic carbon fraction minus individual molecular weight of characterized organic species. This information has been added in lines 496-497, page 17. Line 447-449- The mass of EC contribution to the PBOA factor shown in Figure 11 is 3 times higher than the mass of soil estimated. Then, it is impossible to conclude anything about EC in soil from this data. The reviewer is right and it was not our intention to mention that EC in this factor could be entirely linked to the resuspended dust fraction. We changed the text to be clearer on this point (please see below). Also, we are now more precise in the text and the SI (SI-7a in mass and SI-7b in $\mu$gEC/$\mu$gPM) on the amount of EC in the PBOA factor. Indeed, (as mentioned above) the average contribution calculated for the profile is largely influenced by two values of about 25% for the site in Strasbourg and 18% at Chamonix, but 6 other sites present no EC in this factor (hence the large standard deviation mentioned in Figure 11 of 9 $\pm$7 %). Moreover, the internal variability

of EC according to bootstrap is now presented in SI-7 for all the sites thanks to 100 bootstrap solutions for each site. For the sites of Strasbourg and Chamonix, EC contributions can be quite high but also very uncertain (between 5% - 30% of EC in PBOA mass for Strasbourg for instance). All of this information indicates that the amount of EC is generally less important than the average value presented in the PBOA factor. The text is now changed accordingly, in order to take into account these elements (lines 510-527, page 17): "The large value for the contribution of EC is driven by two high values obtained at the sites of Strasbourg (that reaches 25%) and Chamonix (18%) both influenced by direct and indirect traffic emissions. However, 6 other sites present no EC in PBOA. Moreover, the uncertainties of EC in the PBOA profile of Strasbourg and Chamonix is rather high (between 5 to 30% of PM mass at Strasbourg, see SI-7). On a yearly average, EC apportioned by this factor (0 to 400 ng m-3 depending on the site) is close to the rural EC background in France of about 300 ng m-3 (Golly et al., 2018). This result on the general chemical profile of the PBOA factor tends to infirm the hypothesis of an emission process associating PBOA material with mineral dust resuspension. Indeed, our findings rather suggest that a main part of PBOA (and polyols) are most likely associated with biological particle direct emissions. It leaves only a minor fraction that could be linked to the mechanical resuspension of PBOA with crustal elements. Some minor fraction of EC in this factor could come from resuspended EC-containing dust particles being accumulated in topsoil as demonstrated in previous works (Forbes et al., 2006; Hammes et al., 2007; Zhan et al., 2016). Hence, the origin of the larger fraction of the contribution of EC remains unknown. Our conclusions are in good agreement with those made by Jia and Fraser (2011), based on the concentrations of these chemicals in different types of samples: i.e. size-fractionated (equivalent to PM2.5 and PM10) soil, plant, fungi, atmospheric PM2.5 and PM10. They found that the ambient concentrations of primary saccharide compounds at the suburban site of Higley (USA) are typically dominated by contributions of biological materials rather than resuspension of soil dust particles and associated microbiota."

Line 453 Figure 11- The PBOA factor has an important contribution of EC (ratio of

OC/EC equal to approximately 4.8, similar to values found in secondary organic aerosol formation). Therefore in my opinion this PBOA factor is probably highly contaminated with secondary organic material. That may explain the more than 30 times higher PBOA mass than the mass of polyols. However a more well based evaluation is impossible given the lack of complementary information from the PMF source apportionment. We thank the reviewer for this interesting question. Indeed, it is a major drawback of most of the dozens of PMF studies published every year, based on molecular tracers, that PBOA and Biogenic SOA (BSOA) factors are not distinctly apportioned, considering that a predominant fraction of the organic material is made of modern C in many types of environments in summer, as deduced by 14C studies (Bonvalot et al., 2016; El Haddad et al., 2011; Vlachou et al., 2018).This modern C is part of both the PBOA and the BSOA fractions. It is a strong motivation of this work to define a PBOA factor using molecular tracers to be used in PMF studies. As not being discriminated in a proper PMF factor, BSOA are somehow redistributed into some of the other factors identified by the PMF method. Hence, our PBOA factor may well be a candidate for such a redistribution. However, at least four arguments point against a major influence of BSOA into the obtained PBOA factors: the BSOA loadings should be rather variable across our sampling sites (due to spatial-variation in the emission of primary products, variation in climatology and global sunshine duration, etc.), with a temporal evolution different from that of the PBOA fraction. But, in the present study, we observe a stable ratio of polyols / OCPBOA (3.0±1.5 % on average) over the sites, which does not tend in favor of significant contribution from secondary processes that would introduce more variability in the ratio. Second, the mapping of the PBOA factor in the PMF is rather excellent, as presented now in the PMF description. Indeed for all sites, no "swap" between factors is observed during the bootstrap sensitivity analysis. PBOA is always mapped with itself (see Fig. SI-5), indicating a well identified factor at each site. Third, while the PMF deconvolution of sources is mainly based on co-variation of the concentrations of the chemical species within a given factor, the concentrations of molecular tracers of PBOA (i.e. polyols) and those of a large part of BSOA (like

pinic acid and 3-MBTCA) present very different seasonal cycles (JL Jaffrezo, unpublished data). The work in progress with these data aiming to investigate whether the PBOA source is mixed with SOA is forthcoming. In such context, Srivastava et al. (2018a) have also recently reported quite different seasonal evolution cycles between BSOA (traced thanks to the oxidation products of isoprene ($\alpha$-methylglyceric acid, 2-methylerythritol) and of $\alpha$-pinene (hydroxyglutaric acid)) and PBOA (traced thanks to the polyols). Finally, we further reviewed the literature in order to check if any previous study had investigated the ratio of PBOA tracers to total OC in the PBOA fraction. Indeed, the OC(PBOA)-to-arabitol and OC(PBOA)-to-mannitol ratios in fungal spores were estimated between 7.22–16.25 and 5.20–10.83, respectively (Bauer et al., 2008; Yttri et al., 2011). The OC(PBOA)-to-polyols ratio in our PMF studies is around 16 on average, which is in the range of what is proposed for fungal spores. Overall, even if it is not excluded at this stage that some BSOA fraction may be mixed in our PBOA factor, we still believe that a large contribution is rather unlikely, and that the chemical fingerprint produced by the PMF analyses is representative of the average composition of the PBOA fraction obtained over a large range of sites. We added some text (lines 454-451, page 15) in order to better explain all of this. "This result may be nuanced, in particular during summer, since some extent of mixing between PBOA and Biogenic Secondary Organic Aerosols (BSOA) cannot be entirely excluded. However, several evidences tend in favor of a non-significant mixing between BSOA and PBOA. First, the ratio of polyols-to-OCPBOA shows a low variability from site to site, while it is unlikely that such a secondary process led to the same amount of OC for all sites since they present different meteorology, sunshine duration etc. Second, the bootstrap analysis do not show any "swap" between factors for the PBOA profile for all sites, indicative of a well-defined factor (see Fig. SI-5). Finally, the OCPBOA-to-polyols ratio in this work (about 16) is in the range of ratio expected for fungal spores (12 -27, when arabitol and mannitol are considered together) (Bauer et al., 2008; Yttri et al., 2011)."

References: Bauer, H., Claeys, M., Vermeylen, R., Schueller, E., Weinke, G., Berger, A., and Puxbaum, H.: Arabitol and mannitol as tracers for the quantification of airborne

fungal spores, Atmos. Environ., 42(3), 588–593, 2008. Bonvalot, L., Tuna, T., Fagault, Y., Jaffrezo, J.-L., Jacob, V., Chevrier, F., and Bard, E.: Estimating contributions from biomass burning, fossil fuel combustion, and biogenic carbon to carbonaceous aerosols in the Valley of Chamonix: a dual approach based on radiocarbon and levoglucosan, Atmos. Chem. Phys., 16(21), 13753–13772, 2016. Bozzetti, C., El Haddad, I., Salameh, D., Daellenbach, K. R., Fermo, P., Gonzalez, R., Minguillón, M. C., Iinuma, Y., Poulain, L., Elser, M., Müller, E., Slowik, J. G., Jaffrezo, J.-L., Baltensperger, U., Marchand, N., and Prévôt, A. S. H.: Organic aerosol source apportionment by offline-AMS over a full year in Marseille, Atmos. Chem. Phys., 17(13), 8247–8268, 2017. Canonaco, F., Slowik, J. G., Baltensperger, U., and Prévôt, A. S. H.: Seasonal differences in oxygenated organic aerosol composition: implications for emissions sources and factor analysis, Atmos. Chem. Phys., 15(12), 6993–7002, 2015. Daellenbach, K. R., Stefenelli, G., Bozzetti, C., Vlachou, A., Fermo, P., Gonzalez, R., Piazzalunga, A., Colombi, C., Canonaco, F., Hueglin, C., Kasper-Giebl, A., Jaffrezo, J.-L., Bianchi, F., Slowik, J. G., Baltensperger, U., El-Haddad, I., and Prévôt, A. S. H.: Long-term chemical analysis and organic aerosol source apportionment at nine sites in central Europe: source identification and uncertainty assessment, Atmos. Chem. Phys., 17(21), 13265–13282, 2017. El Haddad, I., Marchand, N., Temime-Roussel, B., Wortham, H., Piot, C., Besombes, J.-L., Baduel, C., Voisin, D., Armengaud, A., and Jaffrezo, J.-L.: Insights into the secondary fraction of the organic aerosol in a Mediterranean urban area: Marseille, Atmos. Chem. Phys., 11(5), 2059–2079, 2011. Forbes, M. S., Raison, R. J., and Skjemstad, J. O.: Formation, transformation and transport of black carbon (charcoal) in terrestrial and aquatic ecosystems, Sci. Total Environ., 370(1), 190–206, 2006.Golly, B., Waked, A., Weber, S., Samake, A., Jacob, V., Conil, S., Rangognio, J., Chrétien, E., Vagnot, M.-P., Robic, P.-Y., Besombes, J.-L., and Jaffrezo, J.-L.: Organic markers and OC source apportionment for seasonal variations of PM2.5 at 5 rural sites in France, Atmos. Environ., 198, 142–157, 2018. Jia, Y. and Fraser, M.: Characterization of Saccharides in Size-fractioned Ambient Particulate Matter and Aerosol Sources: The Contribution of Primary Biological

[Figure]

Aerosol Particles (PBAPs) and Soil to Ambient Particulate Matter, Environ. Sci. Technol., 45(3), 930–936, 2011. Hammes, K., Schmidt Michael W. I., Smernik Ronald J., Currie Lloyd A., Ball William P., Nguyen Thanh H., Louchouarn Patrick, Houel Stephane, Gustafsson Örjan, Elmquist Marie, Cornelissen Gerard, Skjemstad Jan O., Masiello Caroline A., Song Jianzhong, Peng Ping'an, Mitra Siddhartha, Dunn Joshua C., Hatcher Patrick G., Hockaday William C., Smith Dwight M., Hartkopf-Fröder Christoph, Böhmer Axel, Lüer Burkhard, Huebert Barry J., Amelung Wulf, Brodowski Sonja, Huang Lin, Zhang Wendy, Gschwend Philip M., Flores-Cervantes D. Xanat, Largeau Claude, Rouzaud Jean-Noël, Rumpel Cornelia, Guggenberger Georg, Kaiser Klaus, Rodionov Andrei, Gonzalez-Vila Francisco J., Gonzalez-Perez José A., de la Rosa José M., Manning David A. C., López-Capél Elisa., and Ding Luyi: Comparison of quantification methods to measure fire-derived (black/elemental) carbon in soils and sediments using reference materials from soil, water, sediment and the atmosphere, Glob. Biogeochem. Cycles, 21(GB3016), 2007. Sandradewi, J., Prévôt, A. S. H., Szidat, S., Perron, N., Alfarra, M. R., Lanz, V. A., Weingartner, E., and Baltensperger, U.: Using Aerosol Light Absorption Measurements for the Quantitative Determination of Wood Burning and Traffic Emission Contributions to Particulate Matter, Environ. Sci. Technol., 42(9), 3316–3323, 2008. Srivastava, D., Tomaz, S., Favez, O., Lanzafame, G. M., Golly, B., Besombes, J.-L., Alleman, L. Y., Jaffrezo, J.-L., Jacob, V., Perraudin, E., Villenave, E., and Albinet, A.: Speciation of organic fraction does matter for source apportionment. Part 1: A one-year campaign in Grenoble (France), Sci. Total Environ., 624, 1598–1611, 2018a. Srivastava, D., Favez, O., Bonnaire, N., Lucarelli, F., Haef-felin, M., Perraudin, E., Gros, V., Villenave, E., and Albinet, A.: Speciation of organic fractions does matter for aerosol source apportionment. Part 2: Intensive short-term campaign in the Paris area (France), Sci. Total Environ., 634, 267–278, 2018b. Weber S., Uzu G., Calas A., Chevrier F., Besombes JL., Charron A., Salameh D., I Jezek., Mocnik G., and Jaffrezo JL (2018) Attribution of Oxidative Potential fractions to sources of atmospheric PM. Atmos. Chem. Phys., https://doi.org/10.5194/acp-18-9617-2018 Vlachou A., Daellenbach KR., Bozzetti C., Chazeau B., Salazar GA., Szidat S.,

Jaffrezo JL., Hueglin C., Baltensperger U., El Haddad I., and Prévôt ASH (2018) Advanced source apportionment of carbonaceous aerosol by coupling of offline AMS and radiocarbon size segregated measurements over a nearly 2-year period. Atmos. Chem. Phys., https://doi.org/10.5194/acp-18-6187-2018 Yttri, K. E., Simpson, D., Stenström, K., Puxbaum, H., and Svendby, T.: Source apportionment of the carbonaceous aerosol in Norway – quantitative estimates based on 14C, thermal-optical and organic tracer analysis, Atmospheric Chem. Phys. Discuss., 11(3), 7375–7422, 2011. Zhan, C., Zhang, J., Cao, J., Han, Y., Wang, P., Zheng, J., Yao, R., Liu, H., Li, H., and Xiao, W.: Characteristics and Sources of Black Carbon in Atmospheric Dustfall Particles from Huangshi, China, Aerosol Air Qual. Res., 16(9), 2096–2106, 2016.

Please also note the supplement to this comment:
https://www.atmos-chem-phys-discuss.net/acp-2018-773/acp-2018-773-AC1-supplement.pdf

---

## Author Comment (AC2) · 18 Dec 2018

The authors would like to thank Anonymous Referee #4 for his/her review and very useful comments to improve the present paper. We have studied carefully the various comments and tried to answer his/her question point by point in the following discussion.

This is a rich manuscript pooling together datasets from 28 sampling sites across France, focusing on polyols and glucose in the atmosphere. I find the manuscript to fit the scope of the journal, and to be generally well-written. I do have some fairly significant concerns about the analysis and technical comments which should be addressed

prior its publication.

General comments: 1. I find the title misleading, as there is not really use of polyols and glucose as tracers of PBOA. To achieve that, both had to be quantified and recommendations provided on how the formers can be used to estimate the latter. Instead, the manuscript is rather descriptive on polyols and glucose, and results from a largely unclear PMF analysis is given, which leads to my next comment. 2. The PMF analysis is overwhelmingly under-reported and under-explained, given that only its main results are presented. The analysis is actually referred to a report on a url which is no longer valid, or refers to publications in preparation, which is not acceptable, given that one cannot first publish the results and later the analysis. If the authors decide to keep PMF analysis for the revised version of this study, further (and complete) details of the analysis must be provided.

We agree with the reviewer that the PMF methodology was too briefly explained in the first version of the manuscript. Since PMF methodology and results are now very common in the literature, we simply referred to the European FAIRMODE guideline and to our previous papers using some of these data (Waked et al., 2014; Weber et al. 2018). We are also working on another manuscript which should described the used PMF methodology in much more details. However, based on reviewer's comments, the revised version is now presenting in a more extended way the PMF methodology used with this large database, and associated general results. Particularly, we included a more detailed explanation of the method both in the text (please see below) and in the SI (extended text, Fig. SI-5 and Tables SI-3 to 5). Further, discussion on the statistical stability of the PBOA factor, using bootstrap analysis, is now included (Fig. SI-5) and the yearly average contribution of each PMF source-factor at each site is now provided (Fig. SI-4). The url provided for the extended report on the SOURCE program was valid, but pointed to the webpage and not to the pdf. Sorry about that. We changed this reference to point to the pdf instead: report available at https://www.lcsqa.org/system/files/rapport/lcsqa2016-

traitement_harmonise_etude_sources_pmf.pdf Regarding the title accuracy, primary sugar compounds, and in particular polyols (such as arabitol and mannitol) are now commonly recognized as suitable tracers of the PBOA emission sources (Rajput et al., 2018; Zhu et al., 2015, 2016, and therein references). In the present work, we systematically obtained a PBOA profile when concentrations of polyols (defined as the sum of daily arabitol, mannitol and sorbitol concentrations) were included in the datasets used for the PMF analysis. This PBOA profile is characterized by the presence of more than 90% of the total polyols. In this sense, we consider that we use polyols as tracers of the PBOA factor in source apportionment, and that this work is pioneering the definition of a robust chemical profile of such a factor. Further, the chemical profile, thanks to the mass fraction of these tracers, can be used to derive a mass of total PBOA. With all of these, we consider that the title is not misleading and we clarified the main text as follows: Lines 225-240, page 7: "The PMF analysis took advantage of the ME-2 algorithm to add constraints to different chemical profiles (see Tables SI-3 and SI-4 for details). Mainly soft constraints were applied in order to add some prior knowledge about the emission sources and "clean" the different profiles without forcing the model toward an explicit solution. In particular, the polyol concentrations were "pulled up maximally", while levoglucosan and mannosan were set to zero, and EC was "pulled down maximally" in this factor in the PBOA factor. This was achieved to avoid mixing with the biomass burning factor as well as possible influences of unrealistic high contributions of EC to PBOA. Other constraints were added parsimoniously to other factors, targeting specific proxies of sources (Table SI-4). As for the general results of this large PMF study, we identified some well-known sources for almost all the sites (biomass-burning, road traffic, secondary inorganics, dust and sea salt). Two other less-common factors were identified for all sites: secondary biogenic aerosols (probably from marine origin), traced mainly by the presence of MSA, and PBOA, traced by the presence of more than 90% of the polyols total mass in the factor. Table SI-5 and Fig. SI-4 present more detailed description of the chemical tracers in each factor, together with their yearly average contribution for each site, respectively. Hereafter, only the PBOA chemical

profile will be extensively investigated. The uncertainties of this PBOA factor are discussed below and its stability is presented in Fig. SI-5. Bootstrap analysis based on 100 resampling runs evidenced the very high stability of this PBOA factor since the PBOA initial constrained factor was mapped to PBOA bootstrap factor (BF) more than 99% of the time".

Lines 441-446, page 15: "Moreover, the sensitivity of this factor to random noise in the data was investigated thanks to randomly re-sampling the input matrix of observation. In PMF analysis, this is done via the bootstrap method (Paatero et al., 2014) in the constrained run. The PBOA factor was always mapped to itself for 13 of the sites and quasi-always (97%) for the last three ones. It means that the PBOA factor does have a very high statistical stability since it never swaps with another factor (see Fig. SI-5)".

3. Please present your results (e.g. Fig. 6) limited only to PM10 sampling, as it is bound to represent more closely the actual atmospheric concentration, instead of being limited by too low sampling cut-off for the species studied. I recommend maintaining though section 3.3 (PM25/PM10 comparison) to report fine vs coarse mode analysis. We do agree with Anonymous Referee #4. We maintained the section 3.3, reporting and discussing the distribution of ambient polyols (and glucose) concentrations between fine and coarse mode aerosols. Following reviewer's suggestion, we modified the Figures 6 to 8, now limited to PM10 sampling sites only.

4. I find the sampling site denominations used here unsuitable. Urban sites are typically strongly impacted by traffic emissions, so their distinction feels arbitrary. And why rural? Do you mean background? From those denominations it feels like it is lacking filter sampling at forested sites, for example. An improved description of the sampling sites is necessary to better understand its somewhat unexpected results. All of the monitoring sites (except the site of OPE-ANDRA) used in this study are stations of the air quality monitoring networks in France (AASQA). The AASQAs follow well-defined criteria for the classification (typology) of all sites in France. This methodological guide of air quality monitoring is part of the national technical reference. We kept the same

classification. Table SI-1 is also now including a web link for the description of each sampling station as provided by the respective AASQA. In this regard, it should be emphasized that urban sites presented in this study are in fact urban background sites, thus not impacted solely by traffic (in contrast to the roadside sites of Roubaix, Strasbourg, and partly Chamonix). Additionally, the rural sites are localized in rural environments, far enough from any anthropogenic sources (such as traffic, industry, etc.). This information has been added, page 5, lines 145-148.

5. There is certainly a lot to gain from combining several sampling sites, but I find that the analysis has become too shallow, unfortunately. Could you also focus on one sampling site and add more analysis (e.g. comparison with FBAP, total number, other species, wind direction/speed, backtrajectory, etc.) to try to better understand what is driving polyols and glucose atmospheric concentration? The manuscript seems to bring more questions than to answer at this point. Especially when it is kept fairly general (unclear PMF, unclear sampling periods, unclear site characterizations, etc.). We also agree with Anonymous Referee #4 that this paper is only presenting one side of a very large study, the one dealing with average concentrations of a limited number of chemical species for a large number of sites, their seasonal variations, comparison of size distributions, and chemical profile of PBOA factors obtained by source apportionment (SA) studies. This is performed by an ensemble-like study that brings, as mentioned by the reviewer, a different and robust view of the PBOA factor in the atmosphere. We found the term "shallow" a bit restrictive considering the wealth of information that is unearthed and settled by this study. We made the choice of a very different approach than that proposed by the reviewer, who is asking for more classical single site studies. It is clear that the present paper is intended to 1) settle the importance of the PBOA fraction in total PM over a large area, and 2) provide a robust chemical profile of this factor to be used in SA studies. As such, we believe that it can stand alone. However, as now mentioned at the end of the introduction, a second paper in progress will partly address the wishes of the reviewer on the processes behind the introduction of polyols into the atmosphere. And a third one will present some results

of covariation of polyols concentrations and microbial fingerprints in air, soil, and plants at a local scale. We also agree with the reviewer that the source apportionment study was lacking description in the submitted version of this paper, and it is now much more described in this updated version. We do not agree with him / her on the "unclear sampling periods" as Table S1 and Fig. S1 (already present in the submitted version) are clearly giving the dates for the sampling period at each site. We partially agree with him / her on the unclear site description. We updated the sites description table (Table S1) with web links describing each of the station, and now give a reference for the site nomenclature used for French Air Quality network. However, one should keep in mind that the characteristics of the immediate surroundings of the sites are of little importance for the atmospheric concentrations of polyols: this will be described in the second paper, showing for example that the time series of concentrations, for polyols and glucose, are within 10% of each other for 3 sites in the Grenoble area 15 km apart (one downtown in a pedestrian area, one in a park about 2.5 km away, and one in a suburban area 15 km away).

Specific/technical comments: Abstract: Unclear why dust ressuspension would be linked to PBOA factor. We do agree with Anonymous Referee #4 and have rephrased more precisely why dust resuspension could be linked to the PBOA factor. From the current literature, it is not clear if the ambient particulate polyols (tracers of PBOA) enter the atmosphere mainly through biological direct emissions or if they are associated to other materials such as the soil dust particles during resuspension processes. The contribution of crustal materials in PBOA chemical profile can give a good indication of the potential relationships between PBOA factors (polyols) and resuspension of soil dust particles. Our findings evidenced that the mean PBOA chemical profile is clearly dominated by contribution from OM ($78\pm9$ % of the mass of the PBOA PMF factor on average), with only a minor contribution from the dust class ($3\pm4$ %), suggesting that ambient polyols are most likely associated with direct biological particle emissions (e.g. active spore discharge) rather than soil dust resuspension. Our conclusions are in good agreement with those made by Jia and Fraser (2011), based on characterizing

the concentrations of these chemicals in different type of samples: i.e. size-fractionated (equivalent to PM2.5 and PM10) soil, plant, fungi, PM2.5/ PM10. They found that the ambient concentrations of primary saccharide compounds at Higley (USA) are typically dominated by contributions of biological materials rather than resuspension of soil dust particles and associated microbiota.

L.53: PM affects climate, not necessarily negatively. We do agree with Anonymous Referee #4 and have rephrased this sentence (lines 53-54, page 2)

L.57: please refer to a more recent reference for carbonaceous matter. More recent references have been added (line 58, page 2).

L.57-L.66 I suggest focusing on OM on the introduction, rather than OC, an artificial species from analytical limitations. L.63: a significant fraction of OM can be associated with . . . We do agree with Anonymous Referee #4 and we focused on OM (lines 58-65, page 2).

L.72: Please specify in which environments you are referring this figure, including atmospheric layer and aerosol sizes. We rephrased this sentence and information on environment type, aerosol sizes, etc. have been added (lines 73-74, page 3).

L.74-76: And fluorescent techniques? Fluorescent techniques, in particular fluorescent microcopy methods have been previously employed to analyze airborne biological particles. This former technic has the drawback to be laborious and time consuming when it comes to analyze many samples (Bozzetti et al., 2016; Heald and Spracklen, 2009). However, it is worth mentioning that recent fluorescent techniques (e.g. UV-APS, WIBS, etc.) have considerably improve our knowledge about the abundance of airborne biological particles (Fröhlich-Nowoisky et al., 2016; Gosselin et al., 2016; Rajput et al., 2018).

L.79: Unclear how atmospheric transport complements sources and abundances The term "atmospheric transport pathways" has been removed as it can be confusing.

L.101: Datasets Large sets of data have been modified into "datasets" (line 114, page 4).

L.104: Please define atmospheric emission pathway. Do you mean the processes the plant underwent to emit polyols? More precise definition has been added. The term atmospheric "emission pathways" was used to specify how particulate polyols and glucose enter the atmosphere (line 117, page 4).

L.132: Please define "very rural". The OPE-ANDRA site is a rural background site, which lies in the North-East of France, in a crop field area. The term "very rural" has therefore been replaced by a more precise description of OPE-ANDRA site (lines 149-150, page 5).

L.152: Please state that this number typically ranges from 1.2 to 2, so the estimates here represent an upper value of OM, thus a lower estimate of the contribution of PBOA. We do agree with Anonymous Referee #4 and this information has been added (lines 170-172, page 6).

L.185: extra space before comma. The extra space before comma has been removed.

L.186. Define JRC JRC was used to specify the European Joint Research Centre. This definition has been added in the main text (line 206, page 7).

L.194: It is unclear why mix up filter-based BC with already quantified thermo-optical EC. Or there was no EC from DECOMBIO project? Please clarify. Filter analysis for EC-OC and aethalometer measurements were conducted simultaneously for the 3 DE-COMBIO sites within the Arve valley (Chevrier, 2016). Aethalometers give the total BC, thus enabling the decomposition of BC concentrations into its two main constituents: wood-burning BC (BCwb) and fossil-fuel BC (BCff) (Sandradewi et al., 2008). Considering the very specific context of this mountainous valley (with a large influence of meteorology on atmospheric concentrations in winter), the PMF runs with better results in term sof statistical stability and geochemistry were the one with BCwb and

BCff. However, for graphical simplicity, BCwb and BCff are summed up and labelled as EC in the present study. We clarified the main text as follows (lines 211-215, page 7): "The PMF conducted within the DECOMBIO project, for the sites of Marnaz, Chamonix, and Passy, used aethalometer (AE 33) measurements instead of EC (Chevrier, 2016). This complementary measure gives the total black carbon (BC), thus enabling the deconvolution of BC concentrations into its two main constituents: wood-burning BC (BCwb) and fossil-fuel BC (BCff) (Sandradewi et al., 2008). For graphical simplicity, BCwb and BCff were summed up and labeled as EC in the following figures".

L.200: See comment #2 Please see answer at comment #2

L.211: Range values refer to min/max? In terms of readability I prefer you remove this info and present only avg±std. For readability, data has been presented as average ± standard deviation in the revised manuscript, as suggested by reviewers.

L.212: Please define Primary Sugar Compound (SC). The term Primary Sugar Compound (SC) was used to specify polyols together with primary saccharide species. The definition has been added (lines 101-102, page 3).

L.228: Please increase axis font sizes. The axis font sizes have been increased (Fig.2, page 8).

L.233: The asterisk is hard to readily identify. Please show only PM10 cutoff filters on this figure. As explained in comments #3, Fig. 6 to 8 (Fig. S5 as well) are now limited to only PM10 cutoff filters.

L.233: The selected period feels somewhat arbitrary, thus lacking a clear definition of what is shown. Please be more direct on the chosen periods (dd-mm-yyyy) and criteria applied. Selecting a specific date season could indeed be arbitrary. For simplicity, it is quite common to use months rather than the days to calculate seasonal concentration values (Verma et al., 2018). In the present work, seasons were defined as follows: Winter = December to February, Spring = March to May, Summer = June to August,

and Autumn = September to November (see lines 368-370, page 12).

Comments L.255: Please add the information of their estimated atmospheric lifetime. The primary sugar compounds (including polyols and primary saccharide compounds) are actually thought to be relatively stable in the atmosphere (Wang et al., 2018). However, studies investigating their atmospheric lifetime are quite limited. One previous laboratory study has been conducted by the US-EPA to evaluate the stability of these chemicals on filter material exposed to gaseous oxidants as well as in aqueous solutions (simulating clouds and fog droplet chemistry). Findings of this former study have shown that primary sugar compounds remain quite stable up to 7 days (the extent of the testing period), pointing out their suitability for use as tracers of atmospheric transport (Fraser, 2010). This information has added in the updated manuscript version (lines 101-109, page 3).

L.256: It feels like a weak hypothesis to me, from the PBOA perspective, could it be emission ratios change with wind speed, temperature, RH? If focusing on comparable season/meteorology, could the correlation be improved, given distinct emission pathways? And how about interferences from other sources? Is it mixing PM2.5 samples? The reviewer is right that this is a weak hypothesis, but there is very few literature on the subject (and so far no PMF study using glucose concentrations). We removed this sentence and keep working on the topic in order to figure out the sources of atmospheric glucose and its potential relation with the PBOA factor.

L.267: To improve readability, please remove SD and describe only the four average values of both sampling sites, given the interest is the distribution of fine vs coarse mode. Presenting "only" average values may certainly improve the readability. Nonetheless, average $\pm$ standard deviation give an in-depth overview of the statistical size distribution of ambient particulate polyols and glucose concentrations. In this regard, we think that it is preferable to keep these concentration values as Mean $\pm$ SD.

L.290: Please remove "compartment". The term compartment has been removed

L.282: Please indicate the number of samples used on this analysis. The number of samples has been specified (lines 327-328, page 11).

L.301: Does it make sense that PBOA-related polyols are "only" 2-3 times higher in summer in comparison to winter time? The trend behind concentrations in "rural", "urban" or "traffic" feels inconsistent with PBOA interpretation. The seasonal concentrations of polyols presented in this section correspond to the average values over all studied sites. When focusing on one site, the summer concentrations can be 5-8 times higher than those observed in winter, which is consistent with an emission process more likely associated with increased biological activities in summertime. This observation is in good agreement with previous works (Jia et al., 2010a, 2010b; Liang et al., 2016; Verma et al., 2018). Furthermore, one should keep in mind that the values presented are seasonal averages and that daily average concentrations can be much higher in summer. Concerning the second part of the remark (about the trend according to site typology), and to the best of our knowledge, there is no observation in the literature that could support the existence of such an intuitive trend. Conversely, and as mentioned above, our measurements indicate that concentrations are nearly identical for the 3 sites of the Grenoble area, at a scale of c.a. 15 km encompassing several types of sites.

L.404: In which time series? These time series refer to a data in the study conducted by Bonvalot et al., (2016). This sentence has been removed, as the whole paragraph has been modified in the updated manuscript version.

L.440 Please correct sea-salt and not " sea minus salt". This correction has been added

L.445: Unclear sentence. This sentence has been rephrased more precisely (lines 507-524, page 17).

References Bonvalot, L., Tuna, T., Fagault, Y., Jaffrezo, J.-L., Jacob, V., Chevrier, F., and Bard, E.: Estimating contributions from biomass burning, fossil fuel combustion,

and biogenic carbon to carbonaceous aerosols in the Valley of Chamonix: a dual approach based on radiocarbon and levoglucosan, Atmospheric Chem. Phys., 16(21), 13753–13772, 2016. Bozzetti, C., Daellenbach, K. R., Hueglin, C., Fermo, P., Sciare, J., Kasper-Giebl, A., Mazar, Y., Abbaszade, G., El Kazzi, M., Gonzalez, R., Shuster-Meiseles, T., Flasch, M., Wolf, R., Křepelová, A., Canonaco, F., Schnelle-Kreis, J., Slowik, J. G., Zimmermann, R., Rudich, Y., Baltensperger, U., El Haddad, I., and Prévôt, A. S. H.: Size-Resolved Identification, Characterization, and Quantification of Primary Biological Organic Aerosol at a European Rural Site, Environ. Sci. Technol., 50(7), 3425–3434, 2016. Chevrier F (2016) Chauffage au bois et qualité de l'air en Vallée de l'Arve : définition d'un système de surveillance et impact d'une politique de rénovation du parc des appareils anciens. PhD thesis, 239 pp, https://tel.archives-ouvertes.fr/tel-01527559.

[revised manuscript text omitted]

Please also note the supplement to this comment:
https://www.atmos-chem-phys-discuss.net/acp-2018-773/acp-2018-773-AC2-supplement.pdf

---

## Author Response (AR2)

ACP-2018-773

We thank the editors and anonymous referees for their constructive corrections and comments that greatly improved the present manuscript. We have considered carefully the different corrections suggested by both editors and anonymous referees, and made corrections accordingly in the manuscript (*red color*). The detailed answers to the specific questions are given below, point by point *in blue color*.

**Answer to Anonymous Referee #4 comments**

L.47: Please add "organic matter (OM)"

This information has been added (line 51).

L. 48-51: I find the current ending of the abstract somewhat confusing, going into details on a minor contribution of dust. My suggestion would be instead to provide some "useful" results from the analysis, which will certainly increase impact and relevancy of the work. One aspect that comes to my mind would be the estimate of PBOA mass concentration based on polyols (or individual arabitol or mannitol), a number which you can provide with uncertainties based on PMF analysis of 1-year sampling from 16 sites (no small feat). And then finalizing with a well designed sentence on the goal and relevancy of this study would wrap up much better the abstract, in my opinion.

We agree with the reviewer that providing the contribution of Polyols to $PM_{PBOA}$ mass can bring added value. Here we provide some statistics concerning the Polyols-to- $PM_{PBOA}$ ratio. We modified the end of the manuscript by adding the following text (lines 47-49):

*"The Polyols-to-$PM_{PBOA}$ ratio is about 0.024±0.010 on average for all sites, with no clear distinction between traffic, urban or rural typology. Overall, even if the exact origin of the PBOA source is still under investigation, it appears to be an important source of PM, especially during summertime".*

L.64: …"whereas a significant fraction of OM can also be associated with…"
This has been corrected accordingly (line 67).
L.80-81: "…have long been suggested as tracers of PBOA (refs here). For instance…"
Thanks for your attentive review. References are now provided (lines 84-85)

L.91: As the reader may not be familiar with specialized fluorescent based instrumentation, I'd suggest the following replacement: "with real-time detectors of fluorescent PBOA (such as UV-APS and WIBS-3), particularly in rainy periods…"
As also suggested by the editor, we are now providing the definition of both UV-APS and WIBS-3 (lines 94-95).

L.102: remove "actually"
This has been removed (line 105).

L.103: "…(Wang et al., 2018) , although studies…"
Thank for this suggestion, it has been changed accordingly (line 106).

L.117: please remove "(i.e. atmospheric input processes)
This has been removed (line 121).

L.123: Please remove the comma prior "in France".
The comma has been removed.

L.130-133: The reader needs to be guided through the results, and not be "explained to" why some results are not being shown. Unless referring to a companion, such references to future publications do not seem well-suited and I find that your manuscript would be better without it.
We agree with the reviewer and removed the corresponding phrases (lines 133-137).

Figure 4: Does the spread of the slope arabitol vs Mannitol makes more sense if separated by biome (or coarsely by region?).
We have also tried to investigate if sites can be grouped by biome (regionally) or typology when focusing on the spreading of the slope between Arabitol vs Mannitol, but we did not observe any clear pattern.

L.334: please remove "into".
This has been removed.

Section 3.4.1 and so forth: Please include the caveat that different sampling sites do not correspond to the same time period (you could be biased an "urban" site by a rainy summer, for example).
Thank you for this suggestion. We included the following caveat in the main text (lines 362-363)
*"Year of PM sampling campaigns are not concurrent at all sites (see Fig. SI-1)".*

L.413-416: This needs to be rewritten, or removed.
This has been rephrased (line 421).

**Answer to Anonymous Referee #3 comments**

Line 73: change "site" to "sites"
Site has been changed to sites.

Lines 214-216: does not seem correct to label as EC to the sum of BCwb with BCff. Why not BC?
We agree that EC and BC do not refer exactly to the same species. However, this simplification was used to uniform labelling of graphs (i.e. readability and clarity). Moreover, the ratio between ($Bc_{wb}$+$BC_{ff}$) to EC is about 1.4±0.4 for the 3 sites (Chamonix, Marnaz and Passy). So we may overestimate the value given in Figure 10 by a factor 1.4, it may thus indeed explain the higher "EC" values for the 3 alpines valley. Nevertheless, it does not change the general conclusion

Line 176 and others: the tables and figures in the annex are called sometimes as "SI" other times as "S" in the main text and in the annex section. Please make uniform labelling.
The supplemental Tables and Figures are now indexed as SI.

Line 506: I have doubts about the correctness of using Putaud et al (2004b) coefficient (5.6) to estimate the dust fraction. The Figure in Putaud et al (2004b) (Figure 2) from where this coefficient was taken reveals a very large dispersion (R2=0.31) and is highly influenced by one only value. If this sample was removed the coefficient would be much more similar to the value of 15 obtained during the outbreak of Sahara dust. The Ca average fraction in the continental crust gives coefficient values of 28-35 (Mason B. and Moore C. B.: Principles of Geochemistry, 4 Edn., Wiley & Sons, New York, 1982. Wedepohl, K. H.: The composition of the continental crust, Geochim. Cosmochim. Ac., 59, 1217–1232, 1995). Comparison of relative ratio values between Al, Fe, Ca2+ and Ti, in table S6 and in the average crust, shows that in the PBOA factor the proportions of these soil tracers are quite different from the crust, which makes difficult the calculation of a soil contribution in PBOA.

Indeed, the dust concentration may vary with the different reconstruction methods found in the literature (Putaud et al. 2004; Malm et al. 1994; Querol et al. 2002; Perez et al. 2008, or the one you provide). Since we do not have all elements for every sites, we decided to choose Putaud et al, 2004

as a first approximation. Even if we agree that the relationship between dust and Ca seems highly influenced by one extreme point. It seems that without this point, the relationship would be closer to a [dust] = 10 × [Ca], i.e. twice the amount we have, leading to 6% on average for the dust fraction in the PBOA factor, which is still very low.

Besides, we should also be cautious when trying to interpret the other metals in the PBOA source factor since they are potentially associated with important uncertainties. The metals in this factor result in a mixing of different other sources, not only the dust one. It is then expected that the ratio do not fit the one of the average crust.

We decided not to change the value, since it is hard to discuss Putaud et al. without the exact values and detailed explanation of the authors. However, we added the following sentence (lines 496-497):

[revised manuscript text omitted]

---

## Author Response (AR3)

ACP-2018-773

The authors would like to thank the editors for their carefully review. We have considered carefully the different alterations and corrections suggested by editors, and addressed them accordingly in the manuscript (*red color: when text is changed, and blue color when text is removed*). The detailed overview of changes can be accessed in the marked-up manuscript version below.

[revised manuscript text omitted]